# Multicompartmental coacervate-based protocell by spontaneous droplet evaporation

Cheng Qi [1], Xudong Ma[1], Qi Zeng[1], Zhangwei Huang[1], Shanshan Zhang[2], Xiaokang Deng[3], Tiantian Kong [2,4] ✉ & Zhou Liu [3] ✉

Hierarchical compartmentalization, a hallmark of both primitive and modern cells, enables the concentration and isolation of biomolecules, and facilitates spatial organization of biochemical reactions. Coacervate-based compartments can sequester and recruit a large variety of molecules, making it an attractive protocell model. In this work, we report the spontaneous formation of core-shell cell-sized coacervate-based compartments driven by spontaneous evaporation of a sessile droplet on a thin-oil-coated substrate. Our analysis reveals that such far-from-equilibrium architectures arise from multiple, coupled segregative and associative liquid-liquid phase separation, and are stabilized by stagnation points within the evaporating droplet. The formation of stagnation points results from convective capillary flows induced by the maximum evaporation rate at the liquid-liquid-air contact line. This work provides valuable insights into the spontaneous formation and maintenance of hierarchical compartments under non-equilibrium conditions, offering a glimpse into the real-life scenario.

The organizational complexity inherent to biological systems is often rooted in hierarchical structures, where compartmentalization serves as a fundamental element[1-6]. One important mechanism for this compartmentalization is through liquid-liquid phase separation (LLPS), which generates membrane-free coacervate droplets[7,8]. These coacervate droplets are not only key to modern cellular architectures, but also potentially play an important role in the formation of primitive cellular structures—protocells[9,10]. A recent study suggests that natural process like evaporation can concentrate biomolecules to levels that induce LLPS and form coacervate droplets[11]. This is particularly crucial for coacervate-based protocells, as they act as membrane-free compartments capable of concentrating RNA molecules to perform specific functions[7,12-15].

Maintaining the hierarchical structure of coacervate-based protocells is essential for their function as organization hubs[16-20]. To maintain coacervate droplets as distinctive compartments, their fusion

and eventual growth into a bulk condensed phase should be prevented[21,22]. Studies suggest that peptides, proteins, lipids, and even small molecules can adsorb to the coacervate surfaces, stabilizing the droplets and thereby preserving their compartmentalized functions[23-29]. Additionally, nonequilibrium settings (NES), such as physical flows within confined spaces, can also support the maintenance of these compartmentalized structures[11,30,31]. For instance, thermal gradients in heated rock pores containing gas bubbles contribute to the fusion and division of coacervate droplets, resulting in coacervate protocells with maintained sizes[32]; capillary flows can localize and self-assemble molecules, facilitating compartmentalization and the formation of lipid vesicles and coacervates[11,33,34].

In this work, we reveal that evaporation-induced LLPS within a sessile droplet can lead to multiple hierarchically compartmentalised structures, which can be sustained by a flow-assisted mechanism. This is accomplished by evaporating a sessile droplet on a thin-oil-coated

[1]Guangdong Provincial Key Laboratory of Micro/Nano Optomechatronics Engineering, College of Mechatronics and Control Engineering, Shenzhen University, 518060 Shenzhen, Guangdong, China. [2]Department of Biomedical Engineering, School of Medicine, Shenzhen University, 518000 Shenzhen, Guangdong, China. [3]College of Chemistry and Environmental Engineering, Shenzhen University, 518000 Shenzhen, Guangdong, China. [4]Department of Urology, Inst Translat Med, The First Affiliated Hospital of Shenzhen University, Shenzhen Second People's Hospital, Shenzhen, China. ✉e-mail: ttkong@szu.edu.cn; zhouliu@szu.edu.cn

substrate. The droplet is a mixture of dextran, polyethylene oxide (PEO), lactoferrin and ovalbumin. Upon evaporation, dextran and PEO undergo segregative LLPS, while lactoferrin and ovalbumin with opposite charges are complexed to form coacervates. Our experiments reveal an interesting interplay between flow dynamics and multiple LLPS, leading to the spontaneous emergence of complex, membrane-free, hierarchical compartments (Fig. 1). These compartments are cell-sized (-10 μm) and resemble a cell encapsulating an intracellular organelle. Interestingly, when the sessile droplet is partially covered by the organic species, convective capillary flows form stagnation points within the droplet. This prevents the compartments from contacting each other, hindering their fusion. As such, the creation, spatial formation, and maintenance of hierarchical core-shell compartments occur spontaneously. Furthermore, we show that RNAs can be successfully recruited into coacervate compartments. Our method, which involves the spontaneous evaporation of a droplet on an organics-wetted surface, does not require complex manual interventions or intricate designs. This simple strategy allows us to create and maintain complex cellular architectures under laboratory conditions that mimic real-life scenarios.

## Results

### Formation of core-shell coacervate-based compartments

To realize multiple compartmentalization, both segregative and associative LLPS are triggered during evaporation of a sessile droplet consisting of dextran, PEO, lactoferrin, and ovalbumin. Our findings indicate that formation and maintenance of hierarchically structured coacervate-based compartments can be achieved upon the droplet's spontaneous evaporation on an organics-wetted substrate.

Evaporation triggers segregative and associative LLPS in a stepwise manner, resulting in a complex and hierarchical structure with multiple compartments (Fig. 2a). The evolving droplet exhibits a core-shell morphology, with a rim of closely-packed, cell-sized core-shell compartments emerging in the peripheral area of the core region. These compartments, resembling the structure of a protocell encapsulating an intracellular organelle, are expected to coalesce to minimize surface energy[35,36]. Interestingly, they maintain the size and structure against coalescence without any surface-active substances or manual interventions.

To visualize self-arrangement of the polymers and proteins in the sessile droplet during evaporation, we first fluorescently label dextran and PEO, a representative aqueous two-phase system (ATPS) that can undergo segregative LLPS. The fluorescence reveals a PEO-rich central region and a dextran-rich peripheral region in the sessile droplet (Fig. 2b). The densely-packed satellite droplets at the edge of the core region appear to be dextran-rich droplets. These droplets likely emerge from LLPS of the PEO-rich core droplet. Previous experiments showed that an ATPS droplet could undergo multiple segregative

LLPS, creating a morphology of multiple concentric circles[37]. These dextran-rich satellite droplets are expected to merge and move towards the center to form a dextran-rich core inside the PEO-rich region. Surprisingly, in our experiment, they remain densely-packed and immobilized, without coalescing or moving towards the center.

To explore the role of proteins in the hierarchical structure of droplets, we label lactoferrin and ovalbumin with fluorescence. The resulting fluorescent images and intensity distributions demonstrate that both proteins spatially overlap with dextran (Fig. 2c), indicating that they are preferentially partitioned in the dextran-rich phase. Notably, in the peripheral dextran-rich droplet, lactoferrin, ovalbumin and dextran form a single phase, while in dextran-rich satellite droplets, lactoferrin and ovalbumin undergo associative LLPS and phase-separate from dextran. The proteins' complexation prompts the formation of the core dextran-rich small droplets, evidenced by bright-field and fluorescent images (Fig. 2a–c). Both lactoferrin and ovalbumin are significantly enriched in satellite droplets, and the associative LLPS generates a protein coacervate phase and a dextran-rich protein-depleted phase, exhibiting a core-shell structure. As shown in Fig. 2b, the core is crowded and complexed by lactoferrin and ovalbumin. A fluorescence recovery after photobleaching (FRAP) experiment confirms the fluidic property of the core, as shown in Supplementary Figure 1, proving that the core is indeed liquid and thus forms the coacervate.

To study the evolutional compartmentation, we focus on one single core-shell satellite droplet and capture its time-lapse images during evaporation in Fig. 2d. In the first 20 min, segregative LLPS is triggered, forming dextran-rich droplets dispersed in a PEO-rich continuous phase. These droplets tend to fuse to minimize interfacial surface energy (Fig. 2d(i) and (ii)). Then, inside the fused dextran-rich droplet, which can spontaneously recruit proteins (Supplementary Table 1), protein concentrations significantly increase, leading to associative LLPS when coacervates start to appear (Fig. 2d(iii)) and then to fuse (Fig. 2d(iv)). Finally, the core-shell satellite droplet sustains until it dries (Fig. 2d(v)). We find that dextran-rich satellite droplets resulting from segregative LLPS are crucial for forming the core-shell architecture as they can provide crowded compartments to trigger associative LLPS of oppositely charged proteins. Otherwise, the core-shell architecture cannot be formed (Supplementary Fig. 2). Additionally, we observe that the core-shell compartments are still obtainable when the temperature is raised to 60 °C (Supplementary Figure 3), which aligns with hypotheses proposing thermally resilient environments with temperatures around 50–90 °C for primitive cellular structures[38].

### Morphology of ATPS droplet evaporating on organics-wetted and unwetted substrates

The emergence and maintenance of densely packed, cell-sized, core-shell satellite compartments are unexpected and far-from-equilibrium.

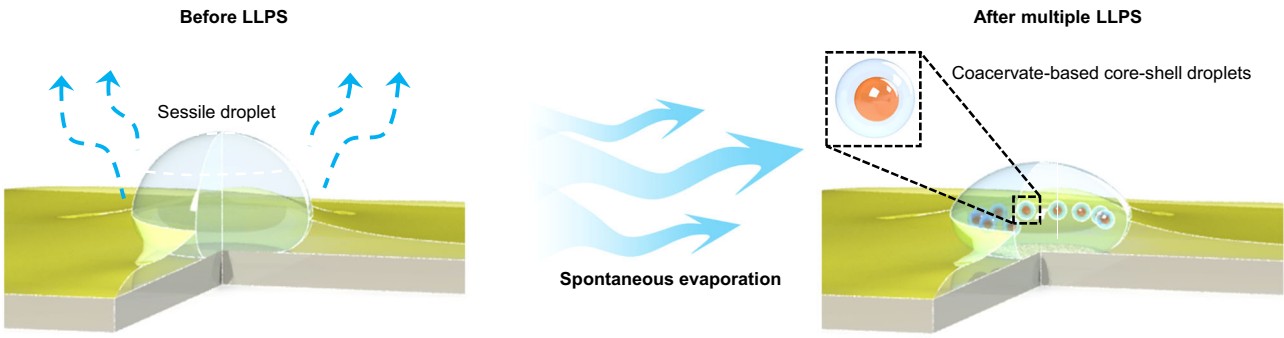

**Fig. 1 | Schematic diagram of a sessile droplet spontaneously evaporating on an organics-wetted surface.** Owing to water loss during evaporation, the concentration of solutes increases significantly, leading to multiple segregative and associative liquid-liquid phase separation (LLPS). As a consequence, a rim of satellite coacervate-based core-shell droplets forms and remains stable under flow dynamics within the evaporating droplet.

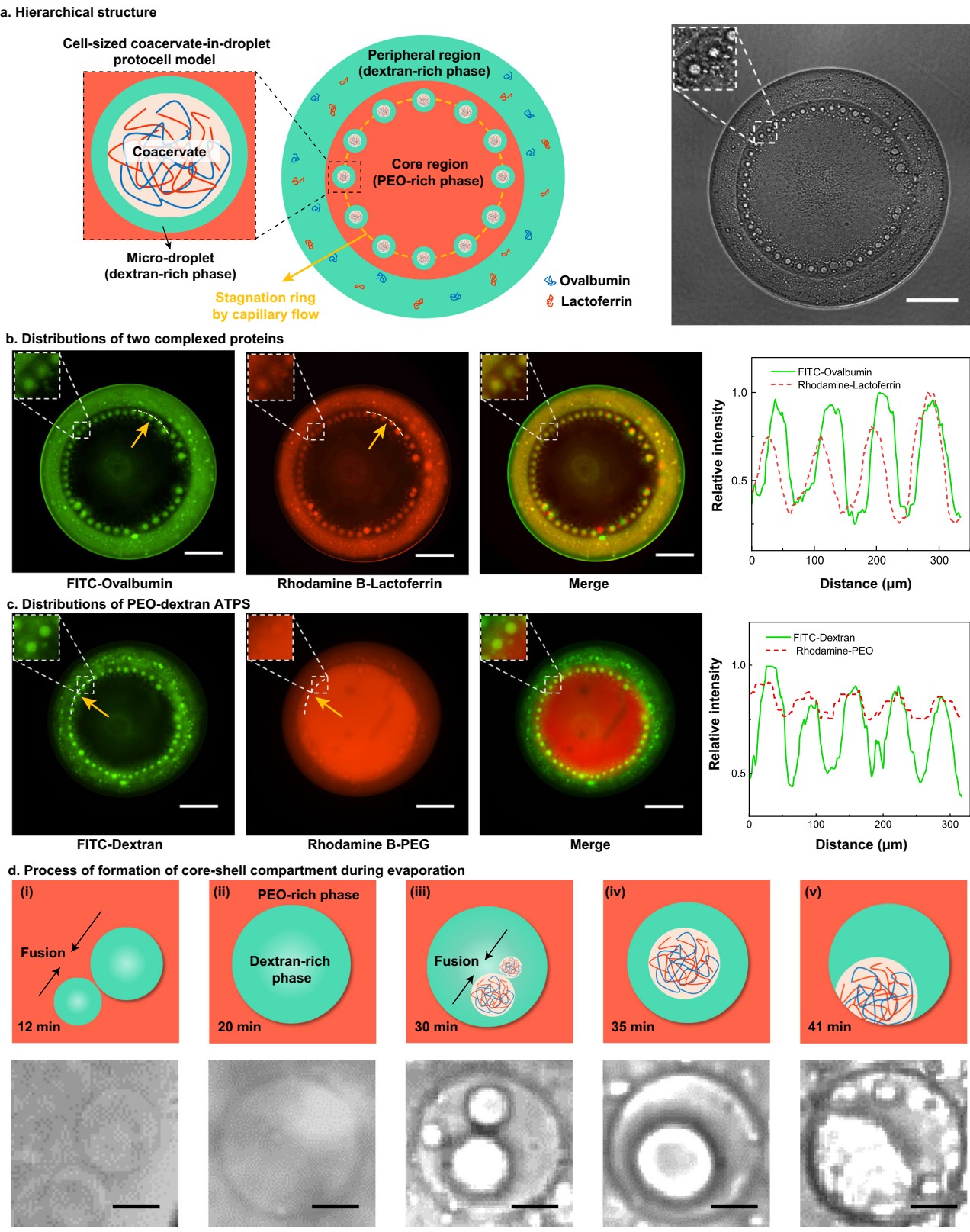

**a. Hierarchical structure**

**b. Distributions of two complexed proteins**

FITC-Ovalbumin | Rhodamine B-Lactoferrin | Merge

**c. Distributions of PEO-dextran ATPS**

FITC-Dextran | Rhodamine B-PEG | Merge

**d. Process of formation of core-shell compartment during evaporation**

(i) 12 min | (ii) 20 min | (iii) 30 min | (iv) 35 min | (v) 41 min

To determine whether lactoferrin and ovalbumin play a vital role, we evaporate a sessile ATPS droplet consisting only of PEO and dextran on a petri dish wetted by organic species (Fig. 3a). Without proteins, the PEO- and dextran-rich phases take up the peripheral and core regions of the droplet, respectively, which is opposite to when proteins are present (Fig. 2). This happens because the PEO-rich phase is more affinitive to the substrate than the dextran-rich phase (without proteins). However, when proteins are present, they partition into the dextran-rich phase that alters its affinity. Indeed, the contact angle measurement demonstrates that proteins make the dextran-rich phase more affinitive to the substrate than the PEO-rich phase, as shown in Supplementary Fig. 4. Consequently, the dextran-rich phase localizes to the peripheral region of the droplet, contrasting with the proteins-free system.

**Fig. 2 | Formation of core-shell compartment during evaporation of a sessile multicomponent droplet and distributions of each component. a** Schematic and bright-field images show the hierarchical structure of an evaporating droplet containing PEO, dextran, and coacervates complexed by two oppositely charged proteins ovalbumin and lactoferrin. **b** Confocal microscopic images show the location of the coacervates (indicated by the overlapping region of FITC-ovalbumin and Rhodamine B-lactoferrin) and the relative fluorescent intensities at the specific locations pointed by the yellow arrow are given. **c** Confocal microscopic images show the locations of PEO-rich and dextran-rich phases and the relative fluorescent intensities at the specific locations pointed by the yellow arrow are given. **d** Schematic and time-serial optical microscopic images show the process of formation of core-shell compartment during evaporation: (i) segregative LLPS gives

rise to dextran-rich droplets dispersed in a PEO-rich continuous phase; (ii) dextran-rich droplets fuse into a larger one; (iii) proteins are recruited by dextran-rich droplet and their concentrations are increased significantly, leading to associative LLPS of proteins and formation of coacervates; (iv) coacervate droplets tend to fuse due to their liquid-like property and it finally forms a stable core-shell compartment; (v) the core-shell compartment remains stable until it is dried. Scale bars are 200 μm in (**a**)–(**c**), and 20 μm in (**d**). In experiments, we used a pair of oppositely charged proteins, lactoferrin and ovalbumin, to form coacervates. The ATPS droplet had the volume of 0.4 μL and contained 3.0 wt% dextran and 1.0 wt% PEO with a volume ratio of 1:1, sitting on 300 μL oil phase (containing 8.0 wt% RSN-0749) in the petri dish. The proteins in the droplets had the concentration of 0.5 wt% before evaporation.

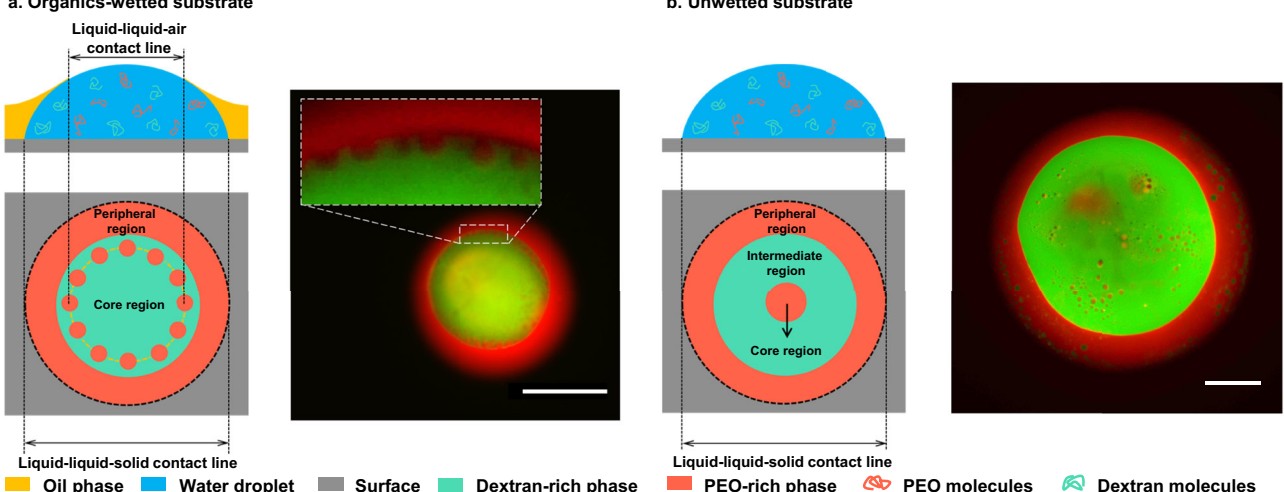

**Fig. 3 | Morphology of an evaporating ATPS droplet (without proteins) resulting from segregative LLPS between PEO and dextran.** Two cases are investigated: a sessile droplet evaporates on (**a**) an organics-wetted substrate, and (**b**) a clean substrate. Schematics and corresponding confocal images of the morphology after evaporation are shown. Dextran and PEO-rich phases are labeled by fluorescein isothiocyanate-dextran (FITC-dextran, green) and rhodamine

B-polyethylene glycol (Rhodamine B-PEG, red), respectively. The sample was prepared by mixing 3.0 wt% dextran and 1.0 wt% PEO with a volume ratio of 9:1. At other volume ratios, a circle of satellite microdroplets was also observed (Supplementary Fig. 6). 0.4 μL of the sample was pipetted onto a petri dish where 300 μL silicone oil was used to wet the substrate a priori in the organics-wetted case. In all samples, 1.0 wt% Pluronic F-68 was included. All scale bars are 200 μm.

The presence of satellite droplets is independent of associative LLPS of oppositely charged proteins. Without lactoferrin and ovalbumin, densely packed cell-sized satellite compartments can still emerge and maintain their sizes at the rim of the core region of the sessile droplet, as shown in Fig. 3a. During evaporation, the ATPS droplet undergoes segregative LLPS and forms the peripheral and core regions, along with a rim of nucleated satellite droplets. Unlike typical phase-separated droplets that grow and merge into one phase, these small droplets do not coalesce, and their sizes maintain until the entire sessile droplet dries. This behavior implies kinetic mechanisms maintaining these far-from-equilibrium droplets.

We propose that the thermodynamic pathway of these small, far-from-equilibrium droplets at the rim of the core region must be arrested. Shum and co-workers studied the kinetics of a sessile droplet of PEO and dextran evaporating on a clean glass surface[31]. They found that small droplets nucleated at the sessile droplet grew quickly, moved towards the center due to Marangoni flow, and coalesced into a large core droplet at the center. We confirm this by replacing the oil-wetted substrate with a clean one and obtain a similar result, where an equilibrium morphology of multilayered concentric circles occurs (Fig. 3b). In contrast, a rim of cell-sized satellite compartments does not form, and we further prove that no core-shell satellite compartment forms when proteins are added into the droplet in this clean-substrate case (Supplementary Fig. 5). The distinct morphologies in Fig. 2 and Supplementary Fig. 5 suggest that the surface property of

the substrate is crucial. We hypothesize that on a substrate wetted by organic species, Marangoni flow must be counteracted by other flows, preventing the small droplets nucleated by LLPS from fusing and moving towards the center of droplet.

## The stagnation ring in flow field of the evaporating sessile droplet

The surface properties of the substrate can affect the behaviors of droplet pinning, evaporation, and drying. In our experiments, organic species on the substrate spreads spontaneously onto the surface of the sessile droplet. Interestingly, we find that the partial coverage of the droplet's surface creates a rim of stagnation points in the flow field. To demonstrate this, we first explore the evaporation of a particle-laden droplet on an organics-wetted substrate. Tracing particles, 0.1 wt% polystyrene (PS) beads with diameter of 1 μm, are added into the droplet, which also contains 1.0 wt% Pluronic F-68 and 10.0 wt% glucose to prevent particle sedimentation. The suspended PS particles are trapped at the rim of deposited droplet, similar to satellite droplets (Fig. 4a). To better understand the formation of the rim of trapped particles, we visualize the interior 3D structure of the evaporating sessile droplet from a side view using a confocal microscope. We use 5 μm purple-fluorescent PS particles, and the oil phase is labeled by blue-fluorescent perylene. Surprisingly, we find that the rim of trapped particles overlaps with the liquid-liquid-air contact line, as seen from the cross-sectional view of B-B in Fig. 4b.

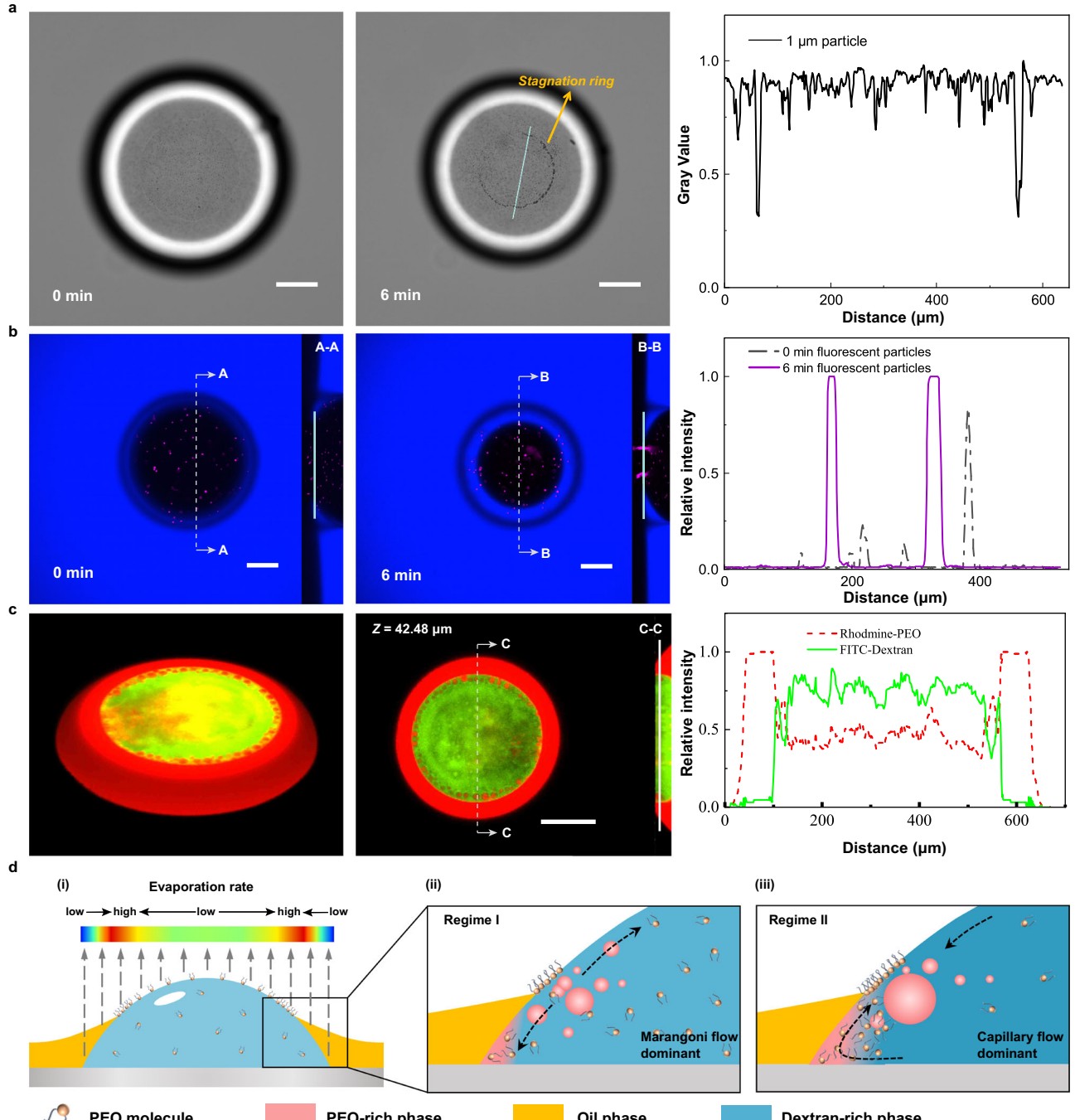

**Fig. 4 | Formation of the stagnation ring and mechanisms of forming and maintaining core-shell satellite compartments by convective capillary flows.** **a** Experimental snapshots showing movement of polystyrene (PS) beads towards the stagnation ring within an evaporating deionized water droplet on an organics-wetted substrate and the corresponding gray value distribution across a blue straight line. **b** Top and side views of distributions of purple-fluorescent PS beads at $t = 0$ min (beginning of evaporation) and $t = 6$ min, as well as the corresponding fluorescent intensities across the blue lines. **c** Three-dimensional reconstruction of an evaporating ATPS droplet, where horizontal sectioning at the height

$z = 35.70$ μm and vertical sectioning in the middle of the droplet are shown. **d** Schematic diagrams to illustrate the mechanism of forming and maintaining the core-shell compartments. In subfigures (**a**) and (**b**), a sessile droplet with a volume of 0.4 μL was placed on a 300 μL oil phase in the petri dish. This droplet contained 1.0 wt% Pluronic F-68 and 10.0 wt% glucose. We used 0.1 wt% purple-fluorescent PS beads as tracing particles, with diameters of 1 μm and 5 μm in subfigures (**a**) and (**b**), respectively. In subfigure (**b**), the oil was labeled by blue-fluorescent perylene. In subfigure (**c**), the composition of the sessile droplet was the same as in Fig. 2a. All scale bars represent 200 μm.

This contact line has the maximum local evaporation rate on the droplet surface (Fig. 4d(i))[39,40]. The water evaporating from the contact line must be replenished by the surrounding liquid, causing convective capillary flows. We believe these flows create stagnation points, which help immobilize the particles and form a stagnation ring near the liquid-liquid-air contact line.

We subsequently investigate whether convective capillary flows prevent satellite droplet growth into a bulk phase. As the sessile droplet phase separates into PEO-rich (red: Rhodamine B-PEG) peripheral and dextran-rich (green: FITC-dextran) core regions, small red fluorescent droplets emerge in the core region near the stagnation ring (Fig. 4c). During the early stages of the evaporation, the PEO

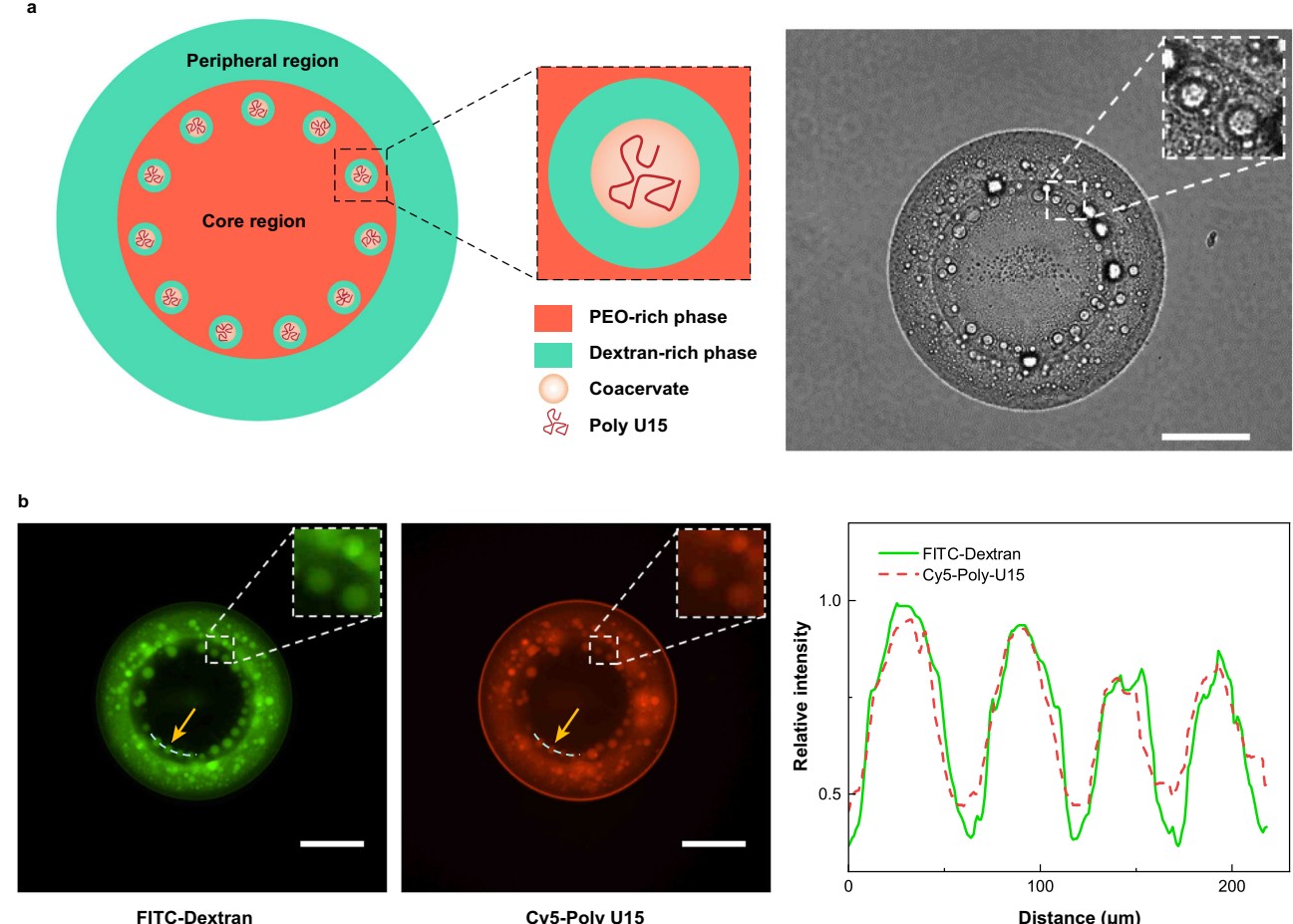

**Fig. 5 | Recruitment of RNA in core-shell compartments indicated by confocal microscopic images. a** Similar core-shell compartments are observed when RNA is added into the multicomponent droplet. **b** Confocal microscopic images show the locations of the core-shell compartment (indicated by the FITC-dextran) and RNA (indicated by the Cy5-Poly U15), and the relative fluorescence intensities of FITC-dextran and Cy5-Poly U15 at the specific locations pointed by the yellow arrow are given. In experiments, the sample was prepared by mixing 3.0 wt% dextran and 1.0 wt% PEO with a volume ratio of 1:1, and was added with 0.5 wt% lactoferrin, 0.5 wt% ovalbumin, and 0.2 mM/L RNA. Then, a 0.4 μL sample droplet was placed on a 300 μL oil phase (containing 8.0 wt% RSN-0749) in the petri dish. The RNA was labeled by Cy5-Poly U15 (red fluorescence). The scale bars are 200 μm.

concentration near the three-phase contact line is higher than the droplet edge due to the maximum evaporation rate at the contact line. The difference in the PEO concentration generates a tension gradient, inducing a Marangoni flow directing from the contact line towards the edge (Fig. 4d(ii)). Simultaneously, the capillary flow caused by water replenishment from the opposite direction counteracts the Marangoni flow.

To determine the dominating flow at different evaporation stages, we compare the relative importance of Marangoni and capillary effects in an evaporating droplet by the dimensionless parameter[41] (Supplementary Discussion 8)

$$K = \left| \frac{\Delta\gamma}{\gamma} \frac{R^2}{h_0 \widetilde{h}} \right| \qquad (1)$$

where $R$ and $h_0$ are the initial radius and height of the sessile droplet, respectively, $\gamma$ is the surface tension of the droplet (about 70 mN/m), $\Delta\gamma$ is the tension difference, and $\widetilde{h}(\ll h_0)$ is the perturbation to the liquid-air interface caused by the internal flow. To evaluate the magnitude of $K$ in the initial stage (regime I), we indirectly measure the interfacial tensions around the three-phase contact line and the droplet edge (method is described in Supplementary Discussion 9). For a sessile droplet ($h_0 \sim 300$ μm, $R \sim 810$ μm) containing 0.031 wt%

PEO, 3.5 wt% dextran and 1.0 wt% Pluronic F-68, after evaporating for 5 min, the interfacial tensions at the contact line and at the edge are 14.70 mN/m and 20.70 mN/m, respectively (see calculation details in Supplementary Discussion 9). The interface perturbation $\widetilde{h}$ is typically very small[41]. In our experiments, this perturbation is too weeny to be observed by the microscope and hence we assume $\widetilde{h} \leq 1$ μm. According to Eq. (1), the magnitude of $K$ is calculated to be larger than 100, indicating that the Marangoni effect dominates over the capillary effect in the initial stage (Regime I, Fig. 4d(ii)). At the later evaporation stage (Regime II), as most PEO molecules move to the peripheral region of the sessile droplet, the surfactant-induced Marangoni effect becomes negligible. Therefore, Marangoni flows are dominant at the early stage, while capillary flows are dominant at the later evaporation stage (Fig. 4d(iii)).

Due to the dominating convective capillary flows at the later evaporation stage, satellite droplets resulting from LLPS can maintain their sizes by avoiding contacting at stagnation points. These dextran-rich, cell-sized satellite droplets can recruit macromolecularly crowded proteins, forming hierarchically core-shell structured compartments where protein coacervates are encapsulated by ATPS droplets. Such natural evaporation-induced hierarchical compartmentalization offers a complex coacervate-based protocell model that mimics a cell encapsulating an intracellular organelle, a phenomenon not previously demonstrated.

### RNA recruitment through membraneless compartmentalization

RNA molecules play critical roles in both the formation and biological functions of primitive and modern cells. The recruitment and accumulation of RNA into compartments can enhance their activities like ribozyme reactions. Complexity in organization and functional coordination is essential for cells. One way to create complexity and diversity is through hierarchical compartmentalization, which forms intracellular organelles for distinctive functions. Evidence suggests that nucleic acids with different lengths and hybridization levels are preferentially recruited into coacervate organelles with distinctive compositions[42]. This preferential partitioning of different types of RNAs in different phases by LLPS allows for the control of RNA distribution and the spatial organization of biochemical reactions in cascades. For example, rRNA (ribosomal RNA) processing and ribosome assembly are coordinated via a core-shell structured organelle[43]. Therefore, the emergence of hierarchical compartmentalization that can recruit and partition different RNAs is of particular importance in understanding modern cell biology.

We further investigate whether RNAs can be actively recruited by hierarchical core-shell compartments that appear after spontaneous evaporation of a sessile droplet. To visualize the spatial distribution of RNAs, we label dextran with FITC (green fluorescence) and a 15-mer oligouridylic acid (U15) with Cy5 (red fluorescence). The addition of RNAs (0.2 mM/L) does not affect the hierarchical morphology of the evaporating droplet, with RNAs mainly distributed in a rim of core-shell satellite droplets near the three-phase contact line (Fig. 5a). The presence of green fluorescent small droplets near the stagnation ring indicates the location of compartments (Fig. 5b). The overlapped fluorescence signals of FITC-dextran and Cy5-U15 across the dashed line (pointed by the yellow arrows) confirm that RNA can accumulate in the hierarchical core-shell compartments, as shown in Fig. 5b.

## Discussion

Hierarchical compartmentalization is essential for controlling and spatially organizing different types of RNAs and protein biochemical reactions in both modern and primitive cells[42,44,45]. This study aims to demonstrate the creation and maintenance of hierarchically structured, coacervate-based compartments through multiple LLPS under laboratory conditions that mimic real-life scenarios. Our findings illustrate how coacervate-based core-shell compartments can form and stabilize during the spontaneous evaporation of a sessile droplet on an organics-wetted substrate. These architectures resemble protocells that encapsulate intracellular organelles.

During evaporation, the sessile droplet undergoes segregative LLPS, where the thermodynamic pathway is kinetically arrested by a stagnation ring in the flow field. This ring forms from convective capillary flows due to the maximum evaporation rate at the liquid-liquid-air contact line. We also show the universality of the stagnation ring phenomenon by hindering evaporation at the droplet's edge using different methods. For example, similar phenomena are observed when a sessile droplet evaporates on a small pit (Supplementary Fig. 9). Thus, far-from-equilibrium droplets of maintained sizes assemble near the stagnation ring. Subsequently, these cell-sized droplets undergo associative LLPS, forming a core-shell structure encapsulating protein-coacervate, similar to a cell with intracellular organelles. Moreover, we demonstrate that these compartments can accumulate RNAs, providing a base for preferential partitioning of various RNA types and their organized biochemical reactions.

Although constructing protocells with hierarchical architectures is possible through sophisticated fabrication procedures in laboratory conditions[46,47], their stabilization strategies remain largely unexplored. By investigating NES evaporation of a sessile droplet on an organics-wetted substrate, we find that a partial coverage of organic phase onto the droplet surface changes the local evaporation rate, creating a rim of stagnation points in the flow field. These stagnation points are crucial for the formation and subsequent stabilization of core-shell, coacervate-based protocells. Our findings reveal the phenomenon resulting from the interplay between spontaneous evaporation and multiple segregative and associative LLPS. Importantly, our approach without sophisticated manual interventions or ingenious designs represents a simple strategy to create and maintain complex cellular architectures under laboratory conditions that mimic real-life scenarios.

## Methods

### Chemicals and solutions

The aqueous two-phase systems (ATPS) were composed of polyethylene oxide (PEO, average Mv ∼ 300,000, power, Macklin) and dextran T500 (Mv ∼ 500,000, BR grade, Ryon). The coacervates were formed by oppositely charged proteins ovalbumin (Mw = 298.38, purity 95%, Shanghai Yuanye) and lactoferrin (purity 95%, Shanghai Yuanye). For fluorescence imaging, the droplet contains 0.015 mg/mL Rhodamine B-PEG (polyethylene glycol, Mw = 40,000, Aladdin), 0.01 mg/mL Rhodamine-lactoferrin (Sangon Biotech, Shanghai, China), 0.015 mg/mL FITC-dextran (average Mw = 500, 000, Sigma-Aldrich), or/and 0.01 mg/mL FITC-ovalbumin (Mw = 298.38, Sangon Biotech, Shanghai, China). For fluorescent RNA, the oligonucleic acid U15, a uridylic acid 15-mer, was obtained from Sangon Biotech and labeled on the 5′-end with Cy5. The U15 oligomer was dissolved in nuclease-free water (20 mM/L), divided into aliquots, and stored at −20 °C. Polystyrene (PS) beads with diameters of 1 μm and 5 μm, and glucose were purchased from Aladdin (Shanghai, China) and BaseLine Chrom Tech Research Centre (Tianjin, China), respectively. Silicone oil PMX-200 (viscosity 100 mPa·s) and perylene (purity 98%) were purchased from Aladdin (Shanghai, China). Pluronic F-68 (10.0 wt%) and Dowsil RSN-0749 RESIN were purchased from Life Technologies Corporation and Dow Chemical Company, respectively. All chemicals are not purified before use.

### Experimental setup

The sessile droplet had a volume of 0.4 uL, sat on a petri dish (polystyrene, Nantong Baiyao Experimental Equipment Co., Ltd.) with a diameter of 35 mm, and evaporated at room temperature (around 23 ± 2 °C) and 50-64% humility. For oil-wetted cases, silicone oil with a volume of 300 μL was coated on the petri dish by spinning and then a water droplet was pipetted on the oil film. As the surface tension of silicone oil is lower than that of water, a water droplet will be completely covered by an oil film. To achieve the partial coverage, we either added water-soluble Pluronic F-68 into the water droplet or oil-soluble RSN-0749 in the oil phase (a detailed discussion is provided in Supplementary Discussion 11). Both bright and fluorescent images of droplet's morphology during evaporation were captured by Nikon ECLIPSE Ti2 microscope. The contact angle of the droplet on the substrate was measured by an inverted Motic AE2000 microscope.

To obtain the relative intensity across a curve in a fluorescent image, the image was imported into the software ImageJ. Segmented line command was used to draw several segmented lines that could coincide with the given curve, and the values of fluorescent intensity across these lines were recorded. These values were then normalized by dividing the maximum fluorescent intensity to obtain the relative intensity.

To illustrate the presence of stagnation points in the flow field in oil-wetted cases, polystyrene (PS) beads with a diameter of 1 μm (about 0.1 wt%) purchased from Aladdin (Shanghai, China) were added into the droplet (0.4 μL) as tracing particles for top-view measurements. To avoid sedimentation of PS particles, 1.0 wt% Pluronic F-68 and 10.0 wt% glucose were added into the water.

To explore the affinity of PEO-rich or dextran-rich phase towards the substrate, the contact angle of the droplet containing each phase resting on the substrate was measured by an inverted Motic AE2000

microscope. To clean the substrate, it was first coated with trichloro (1H, 1H, 2H, 2H-tridecafluoron·octyl) silane (purity > 97% (GC), Aladdin) for 5 s, and then washed continuously with 95% ethanol and deionized water. After that, the substrate was placed inside an oven for drying (at a temperature of about 70 °C) drying for one hour. A sample with a volume about 0.4 μL was pipetted onto the substrate for contact-angle measurement.

## Statistics and reproducibility
Each experiment was independently repeated three times, and consistent results were obtained in Figs. 2a–d, 3a, b, 4a and 5a, b. To ensure reproducibility of results, it is necessary to avoid environmental factors such as vibration and wind that may interfere with an evaporating droplet.

## Reporting summary
Further information on research design is available in the Nature Portfolio Reporting Summary linked to this article.

## Data availability
All data are available within the paper and its Supplementary Information file. Relevant data can be provided by the corresponding author upon request.

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

## Acknowledgements
This work was supported by National Natural Science Foundation of China (Grant Nos. 22322808 (T.K.), 22308219 (C.Q.), 22078197 (Z.L.), 22378267 (Z.L.)), Guangdong Basic and Applied Basic Research Foundation (Grant No. 2023A1515011827 (C.Q.)), Shenzhen Science and Technology Program (RCYX20221008092902010 (Z.L.)) and Research Team Cultivation Program of Shenzhen University (Grant No. 2023QNT018 (T.K.)).

## Author contributions
C.Q., T.K., and Z.L. conceived, designed, and supervised the project. X.M., Q.Z., Z.H., S.Z., and X.D. carried out the experiments. C.Q., T.K., and Z.L. wrote the manuscript. All authors analyzed the data and commented on the paper.

## Competing interests
The authors declare no competing interests.
