## [Peer Review File · Nature Communications]

Multicompartmental coacervate-based protocell by spontaneous droplet evaporationREVIEWER COMMENTS

Reviewer #1 (Remarks to the Author):

Basically, this work shows the drying process of a drop of mixed solution containing dextran, PEO, lactoferrin and ovalbumin on an oil coated surface of petri dish at room temperature. LLPS between dextran and PEO, as well as coacervation between lactoferrin and ovalbumin occurred during the evaporation of water that induced concentration change for all components. The results showed that the coacervate droplets were localised in dextran phase and concentrated towards the interface between dextran and PEO with the evaporation, similar to coffee ring effect.

However, the authors claimed that

‘To create prebiotically plausible nonequilibrium conditions, we pipetted this sessile droplet onto a petri dish wetted by organic species. On early Earth, organophilic silica-rich mineral surfaces were important for preferentially absorbing organic molecules and catalyzing them into prebiotic biochemical polymers.’
‘We have demonstrated that in an organophilic rock pore, the evaporation fluxes at the edge of a sessile droplet are impeded by organic species. As a result, the sessile droplet evolves into membrane-free compartments with a complex hierarchical structure.’

Firstly, the experimental setting is far away from the ‘prebiotically plausible conditions’ as claimed by the authors, that is, oil coated petri dish surface would not create early Earth organophilic silica-rich mineral surfaces. Also, the evaporation effect on flat surface is not same as ‘organophilic rock pore’ as claimed by the authors. So that the whole argument on ‘Our findings provide important insights for the evolution of membraneless, hierarchically-structured, coacervate-based protocells within an organophilic porous rock pores’ is not acceptable.

There is no systematic studies on the effects of concentration, time, and temperature. Furthermore, it is not yet clear how long this structure can be maintained, et al.

Some descriptions in the context seem to come from imagination that does not match the experimental results, and some drawing schemes are incorrect, which can mislead the readers (Figures 1-3).

The author exaggerates their findings in terms of the relationship with protocell or prebiotic conditions. Since LLPS between dextran and PEO, coacervation between oppositely charged species, the sequestration of RNA into coacervate phase, coffee ring effect, are well-known phenomena, this work has not contributed any new concepts to the promotion of research field. Not suitable for publication in Nature Communications.

1. The authors need to be careful with their wording. For example, in line 110-113, ‘Figure 1. (A) Sketch depicting prebiotic environment on early Earth and inset displaying a typical structure of rock surface.’
Is there any reference to support this statement? Otherwise, how does the author know the appearance of rock surfaces on early Earth.

2. Lines 176-181, The scale bar is 100 μm . In experiments, we used a pair of oppositely charged proteins, lactoferrin and ovalbumin, to form coacervates. The ATPS droplet had the volume of 0.4 μL and contained 3.0 wt % dextran and 1.0 wt % PEO with a volume ratio of 1:1, sitting on 300 μL oil phase (containing 8.0 wt % RSN-0749) in the petri dish. The proteins in the droplets had the concentration of 0.5 wt % before evaporation. All scale bars denote 200 μm .

The scales here are confusing (highlighted in yellow). Please clarify.

3. Lines 215-226, 'Figure 3. Morphology of evaporating ATPS droplet resulting from phase separation between PEO and dextran. Three cases were investigated: partially-covered (A), fully-covered (B) and no-covered (C) droplets spontaneously evaporate on a substrate. Corresponding confocal images of the resulting morphology after evaporation are shown in Aii, Bii, and Cii, respectively. The dextran-rich phase is labeled by fluorescein isothiocyanate-dextran (FITC-dextran, green) and PEG-rich phase is labeled by Rhodamine B (red). The scale bar is 100 μm . (A)The droplet had the volume of 0.4 μL and consisted of 4.0 wt % dextran and 0.5 wt % PEO with a volume ratio of 14:1. Pluronic F-68 was added into the droplet and its mass fraction was about 1.0 wt %. (C) We dropped 0.4 μL of the solution (consisting of 3.0 wt % dextran and 1.0 wt % PEO with a volume ratio of 1:1) by micropipettor onto a polystyrene petri dish for observation. All scale bars denote 200 μm .'

Again, the legend in Figure 3 is very confusing. Figures (C), Aii, Bii, and Cii are described, but there are no such labels in the figures.

4. The authors need to carefully revise the English. For example, in line 310, I don't understand what "nuclei acids" supposed to mean

5. The authors mentioned that 'wet-dry cycling'. There was no cycling perform in the paper.

6. Lines 353-354 'To replicate this process, we allowed a compound droplet spontaneously evaporate on a porous mineral substrate that absorbs organic species.' The experiments were performed on petri dish. The description does not match the experiments.

7. The figure labels were inconsistent with the text in figure 4

Reviewer #2 (Remarks to the Author):

I have mixed feelings about this manuscript. On the one hand, the authors demonstrate an original way to produce multi-component coacervate droplets. They have thoroughly documented this novel process, namely controlling evaporation fluxes in an organophilic porous structure, and I find it an interesting and valuable contribution to the literature on engineering structures via liquid-liquid phase separation. On the other hand, I am not convinced by anything they have done that this has anything to do with origins of life, which is the premise on which they base the entire significance of this manuscript. Many routes to multi-compartment droplets in LLPS processes are possible. I see nothing in this work that plausibly argues that this route is more plausible than other routes to multi-compartment structures relevant to pre-biotic genesis of life. The deep question is how does RNA biological chemistry evolve under conditions such as this. The author make a small nod toward this point in their last figure.

I think much of this work could and should be published in Nature Communications. However, I think

straining to make the case that this particular route to LLPS is THE way life evolved is misguided. I would suggest a major re-write that brings out the very interesting features of the new process they have developed. There is no need to insist that THIS is the way life evolved to make the basic work they have done interesting.

This suggests a re-write that emphasizes the interesting work that they have actually done and de-emphasizes the insistence on origin of life significance.

Reviewer #3 (Remarks to the Author):

The manuscript titled “Multicompartmental coacervate-based protocell by spontaneous droplet evaporation” authored by Cheng Qi et. al. describes the formation of core-shell coacervate based protocells driven by physical flows from an evaporating droplet. They use a combination of dissociative phase separation between Dextran and PEO and associative phase separation between lactoferrin and ovalbumin to show structured compartmentalisation. They show that the surface coating of the substrate plays a critical role in the configuration of aqueous two phase systems and the formation of the core shell coacervate structures. They attribute the observed effects to a balance of marangoni effects and capillary effects that drive internal flows within the evaporating droplet.

The work expands on previously published work which show the formation of aqueous two phase droplets within an evaporating droplet by incorporating two additional molecules, lactoferrin and ovalbumin to form internal structures within the dextran, PEO droplets. Currently, the work does not support the conclusions or claims. Please find below for questions about the work.

1. Could the observations of the core-shell coacervates and the general observations for the multicompartment droplet be attributed to the presence of oil? Where the oil mixes with aqueous phase and leads to different multicompartment structures that also give rise to core shell coacervate droplets. Could the oil coat the coacervate droplet and stabilize the droplets to coalescence.
2. Please show evidence that lactoferrin forms coacervates with ovalbumin.
3. Are core-shell structures observed under equilibrium conditions, i.e. at increased concentration? In addition, the zoom in of the images of the core shell structures are hard to see.
4. Why is the ATPS inverted without the proteins?
5. There appears to be co-localisation of RNA into the dextran phase, have the authors checked for overlapping signal?
6. It is not clear how useful the distance vs relative intensity plots are for the narrative.
7. Do the beads show the hydrodynamic flows within the sessile droplet upon evaporation? For example can this be used to confirm the following statement “We believe such capillary flows create stagnation points, which can immobilize the particles and form a stagnation ring near liquid-liquid-air contact line.”
8. The authors say that the balance between marangoni and capillary flows prevent coalescence. Can this be verified by changing the evaporation rate or the size of the droplets which would change the balance

of the flows?

In addition:

Please reference previous work where relevant

We sincerely thank the reviewers for reviewing our work and providing the valuable comments! We have addressed all the comments through further experimentation and have made revisions on the manuscript and supporting information, as specified below. Those revisions are highlighted by **red**.

Response to Reviewer #1:

Comment 1: Basically, this work shows the drying process of a drop of mixed solution containing dextran, PEO, lactoferrin and ovalbumin on an oil coated surface of petri dish at room temperature. LLPS between dextran and PEO, as well as coacervation between lactoferrin and ovalbumin occurred during the evaporation of water that induced concentration change for all components. The results showed that the coacervate droplets were localised in dextran phase and concentrated towards the interface between dextran and PEO with the evaporation, similar to coffee ring effect.

However, the authors claimed that

‘To create prebiotically plausible nonequilibrium conditions, we pipetted this sessile droplet onto a petri dish wetted by organic species. On early Earth, organophilic silica-rich mineral surfaces were important for preferentially absorbing organic molecules and catalyzing them into prebiotic biochemical polymers.’

‘We have demonstrated that in an organophilic rock pore, the evaporation fluxes at the edge of a sessile droplet are impeded by organic species. As a result, the sessile droplet evolves into membrane-free compartments with a complex hierarchical structure.’

Firstly, the experimental setting is far away from the ‘prebiotically plausible conditions’ as claimed by the authors, that is, oil coated petri dish surface would not create early Earth organophilic silica-rich mineral surfaces. Also, the evaporation effect on flat surface is not same as ‘organophilic rock pore’ as claimed by the authors. So that the whole argument on ‘Our findings provide important insights for the evolution of membraneless, hierarchically-structured, coacervate-based protocells within an organophilic porous rock pores’ is not acceptable.

Response 1: The reviewer has raised a reasonable doubt about the validity of using a petri dish wetted by organic species as a proxy for early Earth conditions and whether the outcomes we observed can provide insights to prebiotic processes.

(1) In response to “Firstly, the experimental setting is far away from the ‘prebiotically plausible conditions’ as claimed by the authors, that is, oil coated petri dish surface would not create early Earth organophilic silica-rich mineral surfaces”, we appreciate the reviewer’s point regarding the limitations of our experimental setup. The petri dish coated with organic molecules is indeed a simplified model and not an exact replicate of early Earth conditions. However, it was employed to create an experimental setting where we can rigorously control variables and draw observations that might be analogous to what could occur in nature. The aim was to mimic certain key characteristics of early Earth, such as organophilic surfaces that could concentrate organic molecules and evaporation occurring in wet-dry cycling.

According to the reviewer’s comment, we have acknowledged that our setup is a

simplified approximation that could be improved in future research, and we also revised statements about prebiotic relevance to tune down our arguments in the sections of Abstract, Introduction (Page 2, Line 44-49, Line 53-57, and Line 63-68; Page 3, Line 77-79, and Line 87-89; Page 4, Line 95-96), Results (Page 4, Line 107-112) and Discussion (Page 15, Line 363 and Line 366-367; Page 16, Line 385-395).

(2) In response to “Also, the evaporation effect on flat surface is not same as ‘organophilic rock pore’ as claimed by the authors. So that the whole argument on ‘Our findings provide important insights for the evolution of membraneless, hierarchically-structured, coacervate-based protocells within an organophilic porous rock pores’ is not acceptable”, we appreciate the reviewer's insight and acknowledge that our flat-surface model serves only as a first-order approximation. While it does not capture the full complexity of organophilic rock pores, we argue that it provides valuable insights into key microscale phenomena. The experiments we conducted, specifically focused on microdroplets, are intended to serve as a foundational step toward a more comprehensive understanding of the complex structures within rock pores.

According to the reviewer's comments, we revise the manuscript and replace “porous rock pores” with organics-wetted substrate or substrate wetted by organic species.

Comment 2: There is no systematic studies on the effects of concentration, time, and temperature. Furthermore, it is not yet clear how long this structure can be maintained, et al.

Response 2: We appreciate the reviewer for this helpful comment. To study the effects of concentration, time and temperature on the evaporating process, we have performed supplementary experiments and made relevant discussions.

(1) To investigate the effect of time, we recorded the process of formation of hierarchically core-shell structured compartments at different time points as the sessile droplet evaporates, as shown in the revised Figure 2D. During the first 20 min, segregative LLPS is triggered and forms dextran-rich droplets dispersed in a PEO-rich continuous phase (Figure 2D(i)). These droplets tend to fuse to minimize interfacial surface energy (Figure 2D(ii)). Then inside the fused dextran-rich droplet which can spontaneously recruit proteins, the concentrations of proteins are significantly improved, giving rise to associative LLPS when coacervates start to appear (Figure 2D(iii)) and then to fuse (Figure 2D(iv)). Finally, the core-shell compartment looks like that in Figure 2D(v) at the dried state. By looking into the formation process of core-shell compartments, we find that dextran-rich satellite droplets resulting from segregative LLPS are important for the formation of core-shell architecture as they can provide crowded compartments to trigger associative LLPS of proteins. The above descriptions and discussions of the process are added on Page 6, Line 150-159 in the revised manuscript.

D. Process of formation of core-shell compartment during evaporation

Revised Figure 2D. Schematic and time-serial optical microscopic images showing the process of formation of core-shell compartment during evaporation: (i) segregative LLPS gives rise to dextran-rich droplets dispersed in a PEO-rich continuous phase; (ii) dextran-rich droplets fuse into a larger one; (iii) proteins are recruited by dextran-rich droplet and their concentrations are increased significantly, leading to associative LLPS of proteins and formation of coacervates; (iv) coacervate droplets tend to fuse due to their liquid-like property and it finally forms a stable core-shell compartment; (v) the core-shell compartment remains stable until it is dried.

(2) To investigate the effect of temperature, supplementary experiments are performed and their results are given in Figure S3 (attached below) in the revised supporting information. The final pattern of core-shell compartment is found to be temperature-insensitive when the environment temperature varies from 23 °C to 60 °C. The relevant discussion is provided on Page 6, Line 162-166 in the revised manuscript.

Revised Figure S3. Effects of temperature on formation of core-shell compartments. (A) Schematic diagrams of four stages regarding the formation of core-shell compartment of four stages during evaporation. (B) Duration of each stage as a function of temperature.

(3) To investigate the effect of concentrations, we perform experiments of four

samples with different concentrations of PEO and dextran. The corresponding patterns formed during evaporation are shown in Figure S2 (attached below) in the revised SI. The concentrations tested in these experiments are: (A) 2.0 wt % dextran, 1.0 wt % PEO; (B) 1.5 wt % dextran, 0.5 wt % PEO; (C) 0.5 wt % dextran, 0 wt % PEO; (D) 0 wt % dextran, 0.5 wt % PEO. Rhodamine B (RB)-lactoferrin, RB-PEG, FITC-dextran, FITC-ovalbumin are added for fluorescent visualization. In the absence of PEO or dextran, the core-shell compartment will not be formed and only coacervates are found (Figures S2C and S2D). Only in the case where both PEO and dextran are present, the core-shell compartments can be formed (Figures S2A and S2B). Note that in the sample (A) the concentrations of PEO and dextran are sufficiently high that they already phase-separate before evaporation, which is distinguished from the sample (B). Although the core-shell structure can be formed for the sample (A), the coacervate-core is subtle (zoom-in in Figure S2A(ii)). These findings and implications have been discussed on Page 6, Line 159-162 in the revised manuscript. The experiment details are provided in Section 3 in the revised SI.

Revised Figure S2. Patterns of evaporating sessile droplet consisting of different concentrations of PEO and dextran. Core-shell compartments are formed in (A) and (B), while are absent in (C) and (D).

Comment 3: Some descriptions in the context seem to come from imagination that

does not match the experimental results, and some drawing schemes are incorrect, which can mislead the readers (Figures 1-3).

Response 3: We are grateful to the reviewer's invaluable comment for highlighting the discrepancies between our descriptive text and the experimental results, as well as the inaccuracies in our schematic diagrams. In response to the reviewer's comment, we made following changes in the revised manuscript to ensure accuracy and clarity of our work.

For Figure 1, we revised the schematic diagram to better reflect the actual experimental setup. The new diagram depicts a sessile droplet evaporating on an organics-wetted substrate, instead of a prebiotic environment and porous rock surface.

For Figure 2, we add subfigure (D) that may help readers know the process of formation of core-shell compartments and can illustrate the core-shell structure clearer by zoom-in.

For Figure 3, we improve its resolution to make it clearer.

Comment 4: The author exaggerates their findings in terms of the relationship with protocell or prebiotic conditions. Since LLPS between dextran and PEO, coacervation between oppositely charged species, the sequestration of RNA into coacervate phase, coffee ring effect, are well-known phenomena, this work has not contributed any new concepts to the promotion of research field. Not suitable for publication in Nature Communications.

Response 4: We thank the reviewer for the critical comment. The reviewer is right that liquid-liquid phase separation (LLPS) between dextran and PEO, coacervation between oppositely charged species, RNA sequestration into the coacervate phase, and the coffee ring effect are established phenomena, but our work introduces a significant innovation that we focus on the previously unexplored role of non-equilibrium conditions created by impeded evaporation.

Far from merely aggregating known phenomena, our research aims to understand how they operate under these unique non-equilibrium conditions. We focus on a sessile droplet on an organics-wetted substrate, and show how constrained evaporation rates in this setting give rise to specialized flow dynamics. These dynamics are crucial for stabilizing complex, hierarchically-compartmentalized structures, and they inhibit uncontrolled fusion, thereby providing a robust model of hierarchical protocells.

The novelty of our work is the investigation of how impeded evaporation-induced LLPS can foster the creation and stabilization of complex, membrane-free, hierarchical structures. This emphasis on the role of non-equilibrium conditions, specifically those induced by impeded evaporation, differentiates our research from existing studies, introducing a layer of complexity that has yet to be comprehensively addressed in the literature.

In response to the reviewer's comments, we have clarified these key points in Page 16, Line 386-395 in the revised manuscript. We believe this more accurately addresses the reviewer's concerns while emphasizing the innovative dimensions of our research.

Comment 5: The authors need to be careful with their wording. For example, in line

110-113, 'Figure 1. (A) Sketch depicting prebiotic environment on early Earth and inset displaying a typical structure of rock surface.'

Is there any reference to support this statement? Otherwise, how does the author know the appearance of rock surfaces on early Earth.

Response 5: We thank the reviewer for pointing out the need for accuracy in our wording, specifically regarding the depiction of rock surfaces on early Earth. In response to the reviewer's comment, we changed the relevant wording used in the revised manuscript to improve clarity and accuracy, and we also modified Figure 1 to better reflect the actual experimental setup, where a sessile droplet evaporates on an organics-wetted substrate rather than a "rock" or "porous surface".

Revised Figure 1. Schematic diagram of a sessile droplet spontaneously evaporating on an organics-wetted surface. Owing to water loss during evaporation, solutes' concentrations become sufficiently high, leading to multiple segregative and associative LLPS. As a consequence, a rim of satellite coacervate-based core-shell droplets is formed and remains stable under flow dynamics within the evaporating droplet.

Comment 6: Lines 176-181, The scale bar is 100 μm . In experiments, we used a pair of oppositely charged proteins, lactoferrin and ovalbumin, to form coacervates. The ATPS droplet had the volume of 0.4 μL and contained 3.0 wt % dextran and 1.0 wt % PEO with a volume ratio of 1:1, sitting on 300 μL oil phase (containing 8.0 wt % RSN-0749) in the petri dish. The proteins in the droplets had the concentration of 0.5 wt % before evaporation. All scale bars denote 200 μm . The scales here are confusing (highlighted in yellow). Please clarify.

Response 6: We thank the reviewer very much for pointing out this confusion. We confirm that the scale bars are 200 μm in Figures 2A-C and 20 μm in Figure 2D. They are amended on Page 8, Line 185-186 in the revised manuscript. To ensure uniformity and accuracy, we double-checked the scale bars in all figures and their corresponding descriptions.

Comment 7: Lines 215-226, 'Figure 3. Morphology of evaporating ATPS droplet resulting from phase separation between PEO and dextran. Three cases were investigated: partially-covered (A), fully-covered (B) and no-covered (C) droplets spontaneously evaporate on a substrate. Corresponding confocal images of the resulting morphology after evaporation are shown in Aii, Bii, and Cii, respectively. The dextran-

rich phase is labeled by fluorescein isothiocyanate-dextran (FITC-dextran, green) and PEG-rich phase is labeled by Rhodamine B (red). The scale bar is 100 μm . (A) The droplet had the volume of 0.4 μL and consisted of 4.0 wt % dextran and 0.5 wt % PEO with a volume ratio of 14:1. Pluronic F-68 was added into the droplet and its mass fraction was about 1.0 wt %. (C) We dropped 0.4 μL of the solution (consisting of 3.0 wt % dextran and 1.0 wt % PEO with a volume ratio of 1:1) by micropipettor onto a polystyrene petri dish for observation. All scale bars denote 200 μm .

Again, the legend in Figure 3 is very confusing. Figures (C), Aii, Bii, and Cii are described, but there are no such labels in the figures.

Response 7: We are terribly sorry for not correcting the figure caption after the figure was reorganized. We understand that such discrepancies can make it difficult for the reader to follow the paper effectively. We checked all the figure captions carefully to avoid such mistake. The caption of Figure 3 is revised on Page 10, Line 235-246. We thank the reviewer again for his/her attention to detail, which has helped us in enhancing the quality of our manuscript.

Comment 8: The authors need to carefully revise the English. For example, in line 310, I don't understand what "nuclei acids" supposed to mean.

Response 8: We thank the reviewer for carefully reading our manuscript and pointing out this typo, which is corrected on Page 14, Line 331 in the revised manuscript. We have carefully examined the entire manuscript to avoid such mistakes.

Comment 9: The authors mentioned that 'wet-dry cycling'. There was no cycling perform in the paper.

Response 9: We thank the reviewer for mentioning this concern. In response to the reviewer's comment, we have removed any mention of 'wet-dry cycling' in the revised manuscript and have de-emphasized its relevance to a prebiotic background. We thank the reviewer again for bringing this to our attention, and we believe that these changes improve the clarity and focus of the paper.

Comment 10: Lines 353-354 'To replicate this process, we allowed a compound droplet spontaneously evaporate on a porous mineral substrate that absorbs organic species.' The experiments were performed on petri dish. The description does not match the experiments.

Response 10: We thank the reviewer for pointing out this misunderstanding statement. In the revised manuscript, we amend the manuscript and replace "porous mineral substrate" with "organics-wetted substrate" or "substrate wetted by organic species".

Comment 11: The figure labels were inconsistent with the text in figure 4

Response 11: We thank the reviewer for mentioning these inconsistencies between the figure labels and citations in the context. We redraw the figure 4 and redefine the labels to make sure consistence in the revised manuscript.

Response to Reviewer #2:

Comment 1: I have mixed feelings about this manuscript. On the one hand, the authors demonstrate an original way to produce multi-component coacervate droplets. They have thoroughly documented this novel process, namely controlling evaporation fluxes in an organophilic porous structure, and I find it an interesting and valuable contribution to the literature on engineering structures via liquid-liquid phase separation. On the other hand, I am not convinced by anything they have done that this has anything to do with origins of life, which is the premise on which they base the entire significance of this manuscript. Many routes to multi-compartment droplets in LLPS processes are possible. I see nothing in this work that plausibly argues that this route is more plausible than other routes to multi-compartment structures relevant to pre-biotic genesis of life. The deep question is how does RNA biological chemistry evolve under conditions such as this. The author make a small nod toward this point in their last figure.

Response 1: We appreciate the reviewer's invaluable comments and the recognition of the originality in our work on engineering structures via liquid-liquid phase separation. We understand the reviewer's reservations regarding the implications of our findings for the origins of life. Our original intention is not to assert that our route is the most plausible pathway for the pre-biotic genesis of life. Rather, we aim to explore how complex, membrane-free, multi-compartment structures can be formed on simple natural conditions, instead of sophisticated experimental conditions. The reviewer is also absolutely correct that the crux of the question lies in how RNA biological chemistry could evolve under such conditions. We acknowledge that this is an area that requires deeper investigation. In response to the reviewer's comment, we rewrite to de-emphasizes the insistence on origin of life significance in the revised manuscript.

Comment 2: I think much of this work could and should be published in Nature Communications. However, I think straining to make the case that this particular route to LLPS is THE way life evolved is misguided. I would suggest a major re-write that brings out the very interesting features of the new process they have developed. There is no need to insist that THIS is the way life evolved to make the basic work they have done interesting. This suggests a re-write that emphasizes the interesting work that they have actually done and de-emphasizes the insistence on origin of life significance.

Response 2: We are grateful for the reviewer's constructive feedback and for that the reviewer find the core elements of our work to be of interest and suitable for publication in Nature Communications. We take the reviewer's point about the overemphasis on the origins of life. We agree that our initial focus may have constrained the broader appreciation of the unique features of the liquid-liquid phase separation (LLPS) process we have explored.

In response to the reviewer's suggestion to re-write the manuscript to highlight the novelty of our specific findings—rather than overstating their implications for life's origins, we have undertaken a significant revision of the manuscript highlighted in red.

We have restructured the content to foreground the novel aspects of the controlled evaporation fluxes and the resulting multi-component coacervate droplets. We have

clarified that our work provides a complementary avenue for understanding complex compartmentalization phenomena that could be relevant to various domains, without insisting that this is the definitive mechanism for the origins of life.

We hope that these revisions make the manuscript more aligned with expectations from the reviewer, and we would like to thank the reviewer again for invaluable insights, which have been instrumental in improving the quality and focus of our work.

In summary, in response to the reviewer's suggestion, revisions have been made in the sections of Abstract, Introduction (Page 2, Line 44-49, Line 53-57, and Line 63-68; Page 3, Line 77-79, and Line 87-89; Page 4, Line 95-96), Results (Page 4, Line 107-112) and Discussion (Page 15, Line 363 and Line 366-367; Page 16, Line 385-395).

Response to Reviewer #3:

General comment: The manuscript titled “Multicompartmental coacervate-based protocell by spontaneous droplet evaporation” authored by Cheng Qi et. al. describes the formation of core-shell coacervate based protocells driven by physical flows from an evaporating droplet. They use a combination of dissociative phase separation between Dextran and PEO and associative phase separation between lactoferrin and ovalbumin to show structured compartmentalisation. They show that the surface coating of the substrate plays a critical role in the configuration of aqueous two phase systems and the formation of the core shell coacervate structures. They attribute the observed effects to a balance of marangoni effects and capillary effects that drive internal flows within the evaporating droplet.

The work expands on previously published work which show the formation of aqueous two phase droplets within an evaporating droplet by incorporating two additional molecules, lactoferrin and ovalbumin to form internal structures within the dextran, PEO droplets. Currently, the work does not support the conclusions or claims. Please find below for questions about the work.

Response: We sincerely thank the reviewer for reviewing our work and the valuable comments! We have addressed all the comments through further experimentation and have made revisions on the manuscript, as specified below.

Comment 1: Could the observations of the core-shell coacervates and the general observations for the multicompartment droplet be attributed to the presence of oil? Where the oil mixes with aqueous phase and leads to different multicompartment that also give rise to core shell coacervate droplets. Could the oil coat the coacervate droplet and stabilize the droplets to coalescence.

Response 1: We thank the reviewer for raising these important questions about the role of oil in determining the evaporating pattern. To response the reviewer’s questions, we performed additional experiment where PEO, dextran and two proteins are contained in the sessile droplet in the absence of a wetted oil phase. The flow pattern is shown in Figure S5 (attached blow).

During evaporation, the droplet will first undergo segregative LLPS between PEO and dextran to form a morphology of concentric circles, where the dextran- and PEO-rich phases take up the peripheral and core regions of the sessile droplet, respectively. On their interface, the coacervate satellite droplets are formed due to associative LLPS of complexed proteins. However, no core-shell structure is obtained in this case. Patterns of an evaporating sessile droplet on oil-wetted and unwetted substrates are displayed in Figure 2 and Figure 5S, respectively. By comparing their results, we can conclude that the oil phase gives rise to formation and maintenance of core-shell satellite droplets. We emphasize this point on Page 9, Line 224-228 in the revised manuscript.

A. Distributions of PEO-dextran ATPS

B. Distributions of two complexed proteins

Revised Figure S5. Optical and confocal fluorescent microscopic images showing patterns formed during evaporation of a sessile droplet on a clean substrate and distributions of each component. (A) Confocal microscopic images show the locations of PEO-rich and dextran-rich phases. (B) Confocal microscopic images show the location of the coacervates (indicated by the overlapping region of FITC-ovalbumin and Rhodamine B-lactoferrin).

Comment 2: Please show evidence that lactoferrin forms coacervates with ovalbumin.

Response 2: We appreciate the reviewer for the important comment. In response to the reviewer's comment, we provide additional evidence for the formation of coacervate between lactoferrin and ovalbumin by means of fluorescence recovery after photobleaching (FRAP).

As shown in Figure 2B, the locations of lactoferrin and ovalbumin molecules are overlapped, indicating the complexation between them. In order to further prove that they form coacervate droplets, we thus photobleached the red and green fluorescence in the droplets composed of Rhodamine B-lactoferrin and FITC-ovalbumin in Figures S1A and S1B (attached below), respectively, in the revised SI, and thereafter observed the recovery of the fluorescence in the droplets. Both red and green fluorescence were recovered within 3 minutes. The experiments demonstrated the good fluidity of lactoferrin and ovalbumin molecules within the droplets, which were proved to be liquid-like coacervates. The discussion is now added on Page 6, Line 145-149 in the revised manuscript and the details of FRAP experiment are described in Section S1 in the revised SI.

Revised Figure S1. FRAP experiments of lactoferrin/ovalbumin complex coacervate droplets with fluorescence labelled Rhodamine B-lactoferrin and FITC-ovalbumin. Fluorescent images at different time points of a FRAP experiment of Rhodamine B-lactoferrin in (A) and FITC-ovalbumin in (B), and their corresponding fluorescence recovery curves in (C). The sample is prepared by mixing 10 wt % dextran, 5 wt % ovalbumin and 5 wt % lactoferrin. The proteins ovalbumin and lactoferrin are labelled with Rhodamine B-lactoferrin (red fluorescence) and FITC-lactoferrin (green fluorescence), respectively. Scale bars are 50 μm .

Comment 3: Are core-shell structures observed under equilibrium conditions, i.e. at increased concentration? In addition, the zoom in of the images of the core shell structures are hard to see.

Response 3: We thank the reviewer for raising this important question. In a response, we performed an experiment of an evaporating droplet consisting of 2.0 wt % dextran, 1.0 wt % PEO. At these high concentrations, dextran and PEO already phase-separate before evaporation. We observed a similar core-shell structure. It suggests that these core-shell compartments can be still produced and maintained under equilibrium conditions. This new result is added in Section 3 (Figure S2A) in the revised SI. In addition, we magnify the core-shell structure in Figure 2D in the revised manuscript. The structure as well as the process of its formation is now clear.

These redrawn figures are attached below:

Figure S2A. Confocal microscopic images showing patterns of evaporating sessile droplet consisting of different concentrations of PEO and dextran. Core-shell compartments are formed in (A).

D. Process of formation of core-shell compartment during evaporation

Revised Figure 2D. Schematic and time-serial optical microscopic images showing the process of formation of core-shell compartment during evaporation: (i) segregative LLPS gives rise to dextran-rich droplets dispersed in a PEO-rich continuous phase; (ii) dextran-rich droplets fuse into a larger one; (iii) proteins are recruited by dextran-rich droplet and their concentrations are increased significantly, leading to associative LLPS of proteins and formation of coacervates; (iv) coacervate droplets tend to fuse due to their liquid-like property and it finally forms a stable core-shell compartment; (v) the core-shell compartment remains stable until it is dried.

Comment 4: Why is the ATPS inverted without the proteins?

Response 4: We thank the reviewer for this comment. In the revised manuscript, we make explanation for the phenomenon in more details.

As the reviewer pointed out, the phenomena were distinct in the presence and absence of the two proteins (lactoferrin and ovalbumin). That is, in the absence of proteins (namely, a pure ATPS sessile droplet), evaporation will lead to LLPS of PEO- and dextran-rich phases, which take up the peripheral and core regions of the sessile droplet, respectively. It is consistent with the phenomenon observed in the study by Shum *et al.* (*Nat. Commun.* 12 (2021) 3194). The phenomenon is attributed to the fact that PEO-rich phase is more affinitive to the substrate than the dextran-rich phase. Therefore, the PEO-rich phase tends to take up the peripheral region of the sessile droplet. However, in the presence of proteins, the proteins partitioned in the dextran-rich phase alter its affinity towards the substrate. As indicated by the contact angle measurement in Figure S4, addition of proteins makes the dextran-rich phase more affinitive to the substrate than the PEO-rich phase, and hence the dextran-rich phase takes up the peripheral region of the sessile droplet, which is opposite to the case in the absence of proteins.

We have explained the distinct phenomena in more details on Page 8, Line 197-206 in the revised manuscript.

Comment 5: There appears to be co-localisation of RNA into the dextran phase, have the authors checked for overlapping signal?

Response 5: We thank the reviewer for this helpful suggestion. According to the suggestion, we checked the overlapping signal of RNA and FITC-dextran as shown in

the below figure, which indicates a high degree of overlap between the two fluorescence signals. It illustrates that RNA is indeed partitioned in the core-shell structured compartments, which may serve as a protocell model in the RNA world hypothesis.

Figure. Confocal microscopic images show the locations of the core-shell compartment (indicated by the FITC-dextran) and RNA (indicated by the Cy5-Poly U15)

Comment 6: It is not clear how useful the distance vs relative intensity plots are for the narrative.

Response 6: We appreciate the reviewer’s comment about the utility of the distance vs. relative intensity plots presented in Figure 5B. The aim of these plots is to provide a quantitative assessment supporting our qualitative observations that RNA was effectively recruited into the dextran-rich compartments. By plotting relative intensity against distance, we demonstrate that the fluorescence signals from RNA and dextran overlap considerably, which indicates successful recruitment of RNA into these compartments.

Comment 7: Do the beads show the hydrodynamic flows within the sessile droplet upon evaporation? For example can this be used to confirm the following statement “We believe such capillary flows create stagnation points, which can immobilize the particles and form a stagnation ring near liquid-liquid-air contact line.”

Response 7: We thank the reviewer for raising a comment we need to clarify. In experiments, the beads are used to show the hydrodynamic flows within the sessile droplet under evaporation. In Figures 4A and 4B, we compared distribution of beads at $t = 0$ and $t = 6$ min. At the beginning of evaporation ($t = 0$), the beads are uniformly distributed within the sessile droplet, while after 6 mins past the beads moved towards a rim. We further confirmed that in Figure 4B this rim was located at the liquid-liquid-air contact line. As is well known, the local evaporation rate at the contact line is maximum such that water evaporating from the contact line is replenished by surrounding water, giving rise to convective capillary flows. Therefore, we concluded: “such capillary flows create stagnation points, which can immobilize the particles and form a stagnation ring near liquid-liquid-air contact line”.

Comment 8: The authors say that the balance between maragoni and capillary flows prevent coalescence. Can this be verified by changing the evaporation rate or the size

of the droplets which would change the balance of the flows?

Response 8: We thank the reviewer for this important question. We recognize that some statements and Figure 4 may not be unclear enough. The reason of non-coalescence of those core-shell compartments is the presence of *convective* capillary flow. We apologize for any confusion caused and appreciate the reviewer’s diligence in bringing this to our attention. Figure 3 can be used to verify whether the convective capillary flows can prevent coalescence. As depicted in Figure 4D(iii), these convective capillary flows are induced by the presence of a wetted oil phase. By comparing Figures 3A and 3B, it is found that red PEO-rich satellite droplets, which are arrayed at a rim in Figure 3A, are moved towards the edge of the sessile droplet and coalesce into a bulk peripheral phase in Figure 3B, known as the coffee ring effect.

In response to the reviewer’s comment, we modify the statement on Page 11, Line 269-270, and the schematic diagrams in Figures 4D(ii) and 4D(iii) in the revised manuscript, where in Regime II a “convective” capillary flow can be visualized in Figure 4D(iii).

Revised Figure 4D. Schematic diagrams to illustrate the mechanism of forming the core-shell structured compartments.

Besides inducing convective capillary flows by the wetted oil, evaporation of a sessile droplet on a small pit can also induce convective capillary flows to form and maintain core-shell compartments (Figure S8 in the revised manuscript). These revisions not only clarify the mechanism but also add depth by showing the multiple ways in which convective capillary flows can be induced.

Revised Figure S8. Evaporation of droplet on a small pit. Core-shell satellite droplets be observed on the zoom-in region.

Comment 9: Please reference previous work where relevant

Response 9: We thank the reviewer for his/her suggestion to reference previous relevant studies. We carefully address this concern by incorporating appropriate references into the revised manuscript. The newly cited references are listed below:

1. We made a citation of Refs. 1 and 2 after the statement “The organizational complexity inherent to biological systems is often rooted in hierarchical structures, where compartmentalization serves as a foundational element” on Page 2, Line 37-39.

2. We made a citation of Ref. 10 after the statement “These coacervated droplets are not only key to modern cellular architectures, but also hypothesized to have aided in the formation of primitive cellular structures—protocells” on Page 2, Line 42-44.

3. We made a citation of Ref. 25 after the statement “Studies suggest that peptides, proteins, lipids, and even small molecules can also adsorb at the surface of coacervates, preventing their uncontrolled fusion and thereby preserving their compartmentalized functions” on Page 2, Line 53-55.

4. We made a citation of Ref. 30 after the statement “capillary flows have been shown to localize and self-assemble molecules for compartmentalization, and subsequently facilitate the formation of lipid vesicles and coacervates” on Page 2, Line 60-62.

5. We made a citation of Ref. 33 after the statement “Previous experiment has shown that ATPS droplet can undergo multiple LLPS, giving rise to a morphology of multiple concentric circles.” on Page 5, Line 128-133.

6. We made a citation of Ref. 43 after the statement “Although construction of protocells with hierarchical architectures is possible through sophisticated fabrication procedures in laboratory conditions” on Page 16, Line 384-385.

REVIEWER COMMENTS

Reviewer #1 (Remarks to the Author):

It is appreciated that the authors made such an effort to perform some additional experiments and revise the manuscript accordingly. The paper looks in a better shape than the previous version. However, there are still major issues that need to be addressed.

1) Lines 187-189, 'The ATPS droplet had the volume of 0.4 μ L and contained 3.0 wt % dextran and 1.0 wt % PEO with a volume ratio of 1:1, sitting on 300 μ L oil phase (containing 8.0 wt % RSN-0749) in the petri dish.'

Please explain why RSN-0749 was added and what role it plays. Was it used in all experiments? The Data Sheet of DOWSIL™ RSN-0749 Resin showed that is a blend of approximately 50 percent trimethylsiloxysilicate and 50 percent cyclopentasiloxane. After the evaporation of the cyclopentasiloxane, the high-molecular-weight silicone resin forms a highly durable film.

2) Why is the background of Figure 2B (left) green.

3) Figure 3 does not make sense at all. It seems to compare 'Morphology of evaporating ATPS droplet on organo- and hydrophilic substrate'. However, the two substrates used are '(A) an organics-wetted surface or (B) a clean surface (petri dish)'. Firstly, polystyrene petri dish cannot be considered as hydrophilic substrate. Secondly, why is the composition of the droplets different in each case? '(A) The droplet had the volume of 0.4 μ L that consisted of 4.0 wt % dextran and 0.5 wt % PEO with a volume ratio of 14:1, and 1.0 wt % Pluronic F-68. 300 μ L silicone oil was used to wet the substrate a priori. (B) We dropped 0.4 μ L of the solution (consisting of 3.0 wt % dextran and 1.0 wt % PEO with a volume ratio of 1:1) by micropipettor onto a petri dish for observation.' If the aim was to compare the effect of substrate properties on morphology, the droplets should be kept the same. Otherwise, it can't be sure what affected the morphology. Thirdly, what was the role of Pluronic F-68 in case (A)? Pluronic F-68 is a triblock copolymer with amphiphilic property that is often used as solubilizers and emulsifiers similar to surfactants. With the participation of amphiphilic molecules, the mechanism of droplet formation may change dramatically.

4) Please clearly indicate which experiment used RSN-0749 or Pluronic F-68 and explain the reasons for using these additives.

5) It is also noted that the composition of the droplets (the concentration and the volume ratio of dextran to PEO) were different in each Figure. Please explain why.

Reviewer #2 (Remarks to the Author):

The authors have addressed my principal concern with the first submission. I believe that this version is suitable for publication in Nature Communications.

Reviewer #3 (Remarks to the Author):

The manuscript titled “Multicompartmental coacervate-based protocell by spontaneous droplet evaporation” authored by Cheng Qi et. al. describes the formation of core-shell coacervate based protocells driven by physical flows from an evaporating droplet.

The authors have undertaken additional experiments to address the questions that I had about their previous manuscript.

The additional experiments have provided more evidence to support their arguments. There are still some remaining points:

1. With the additional control experiments they show that a major factor for preventing droplet coalescence is due to the oil. This suggests that the balance of flows driven by evaporation of the droplet is not the dominating factor for the stabilizing the core shell coacervate structures. That the stabilization is a thermodynamic rather than a kinetic effect. If this is the case, the manuscript should be revised accordingly to:
 - More clearly describe the “organics-wetted surface” in the abstract and introduction (for example lines 64-68)
 - Clearly describe the factors which are important for the droplet stabilization
 - Describe the context of the maragoni and capillary effects in light of the new experiments.
2. The authors say that the added value of their work is that they show the formation of hierarchical structures in “natural conditions”. It can be argued that a simulated surface with oil on it is not natural and is a laboratory mimic for a potential real-life scenario. As such, it does not set this work above other studies which show hierarchical structuration in coacervates in laboratory scenarios. The text should be modified to focus on the phenomena that drives and stabilizes the droplets rather than the “natural conditions”.
3. Line 96. They say that this work shows the “potential origins of complex architectural structures under natural conditions”. This sentence should be justified in light of a number of papers which show the formation of multistructures under different conditions.
4. Language can be improved in various places.

Response to Reviewer #1:

Comment 1: It is appreciated that the authors made such an effort to perform some additional experiments and revise the manuscript accordingly. The paper looks in a better shape than the previous version. However, there are still major issues that need to be addressed.

Response: We sincerely appreciate the reviewers for his/her time and valuable feedback on our work. We have addressed all the comments through additional experiments and have revised the manuscript and supporting information accordingly. These changes are highlighted by red.

Comment 2: Lines 187-189, 'The ATPS droplet had the volume of 0.4 μ L and contained 3.0 wt % dextran and 1.0 wt % PEO with a volume ratio of 1:1, sitting on 300 μ L oil phase (containing 8.0 wt % RSN-0749) in the petri dish.' Please explain why RSN-0749 was added and what role it plays. Was it used in all experiments? The Data Sheet of DOWSIL™ RSN-0749 Resin showed that is a blend of approximately 50 percent trimethylsiloxysilicate and 50 percent cyclopentasiloxane. After the evaporation of the cyclopentasiloxane, the high-molecular-weight silicone resin forms a highly durable film.

Response: We thank the reviewer for raising a confusion about the role of RSN-0749 that we need to clarify. The reviewer also inquired about the role of Pluronic F-68 in comment 4 and 5. These additives perform similar functions; they were both used to control the degree of oil film coverage on a sessile droplet. Only when the sessile droplet is partially covered by the oil phase, stagnation points, denoted by "stagnation ring" in Figure 4A, can appear in the flow field which can result in a circle of satellite phase-separated microdroplets. The roles of both Pluronic F-68 and RSN-0749 are detailed below.

Explanation of roles of Pluronic F-68 and RSN-0749

The evaporation of sessile water droplets on an organics-wetted substrate reveals that a partial oil film coverage is crucial for forming a circle of stagnation points near the water-oil-air contact line. This partial coverage was experimentally achieved by either introducing water-soluble Pluronic F-68 into the water droplet or incorporating oil-soluble RSN-0749 into the oil phase.

To explore the effects of Pluronic F-68 and RSN-0749 on the coverage morphology, we pipetted a droplet onto a silicon oil-wetted substrate and observed the 3D reconstructed architecture with a confocal microscope. For fluorescent visualization, rhodamine B isothiocyanate-dextran (RBITC-dextran, red fluorescence) and perylene (blue fluorescence) were added to the the water droplet and oil film, respectively.

Three distinct scenarios were investigated: (A) a water droplet containing F-68 on a pure oil-wetted substrate; (B) a pure water droplet on an RSN-0749 laden oil-wetted substrate; (C) a pure water droplet on a pure oil-wetted substrate. The corresponding morphologies are shown in Figure S10 of the revised supporting information (SI). In scenarios (A) and (B), the droplet exhibited partial coverage, while in scenario (C), it

was completely engulfed by the oil film, as indicated by yellow arrows in Figure S10C.

Figure S10 (in the revised SI). Coverage morphology of a water sessile droplet on a silicon oil wetted substrate. Pluronic F-68 (1.0 wt %) and RSN-0749 (8.0 wt %) are added to the water droplet (A) and oil phase (B), respectively. There are no additives in (C) which acts as a negative control. The water and oil phases are fluorescently labeled with RBITC-dextran (red) and perylene (blue), respectively. In (C), the oil film climbs onto the top of the sessile droplet, as denoted by yellow arrows.

Both Pluronic F-68 and RSN-0749 can reduce the coverage of oil film. Comparative analysis of Figures S10A and S10B suggests that Pluronic F-68 is more effective, evidenced by minimal oil wetting on the droplet's surface in Figure S10B(ii). However, Pluronic F-68 could interfere with proteins in LLPS droplet during evaporation. Therefore, for experiments involving proteins within the droplet, as shown in Figures 2 and 5, RSN-0749 was used. In other experiments (Figures 3 and 4), we used Pluronic F-68.

Without either RSN-0749 or Pluronic F-68, the droplet becomes fully covered by the oil film, due to the lower surface tension of silicone oil compared to water. In this scenario, the stagnation ring is absent and immobilization effect disappears; thus, microdroplets tended to coalesce into a bulk phase rather than forming a circle. Eventually, a Janus-like structure of droplet is produced (Figure S11 in the revised SI).

To address the reviewer's comments, we have included the above detailed discussion in the Section S11 of the revised SI. The concise explanation of the roles of RSN-0749 and Pluronic F-68 are provided on Page 17, Line 419-423 in the revised manuscript.

Figure S11 (in the revised SI). Morphology of an evaporating ATPS droplet on an oil-wetted substrate resulting from segregative LLPS between PEO and dextran. The droplet contains 3.0 wt % dextran and 1.0 wt % PEO with a volume ratio of 1:1. Neither Pluronic F-68 nor RSN-0749 is added. Scale bars are 100 μm .

Comment 3: Why is the background of Figure 2B (left) green.

Response: We thank the review for pointing out this issue. The green background in Figure 2B has been corrected in the revised manuscript.

Comment 4: Figure 3 does not make sense at all. It seems to compare ‘Morphology of evaporating ATPS droplet on organo- and hydrophilic substrate’. However, the two substrates used are ‘(A) an organics-wetted surface or (B) a clean surface (petri dish).’ Firstly, polystyrene petri dish cannot be considered as hydrophilic substrate. Secondly, why is the composition of the droplets different in each case? ‘(A) The droplet had the volume of 0.4 μL that consisted of 4.0 wt % dextran and 0.5 wt % PEO with a volume ratio of 14:1, and 1.0 wt % Pluronic F-68. 300 μL silicone oil was used to wet the substrate a priori. (B) We dropped 0.4 μL of the solution (consisting of 3.0 wt % dextran and 1.0 wt % PEO with a volume ratio of 1:1) by micropipettor onto a petri dish for observation.’ If the aim was to compare the effect of substrate properties on morphology, the droplets should be kept the same. Otherwise, it can’t be sure what affected the morphology. Thirdly, what was the role of Pluronic F-68 in case (A)? Pluronic F-68 is a triblock copolymer with amphiphilic property that is often used as solubilizers and emulsifiers similar to surfactants. With the participation of amphiphilic molecules, the mechanism of droplet formation may change dramatically.

Response: We thank the reviewer for the pointing out the important issues that requires clarification.

(1) First, the reviewer is that that hydrophilic substrate is inappropriate wording for polystyrene petri dish. Follow the reviewer’s suggestion, we have changed the title to “Morphology of ATPS droplet evaporating on organics-wetted and unwetted substrates” on Page 7, Line 177-178 of revised manuscript.

(2) In response to different the composition of the droplets, we appreciate the reviewer's help comment and acknowledge the need to better organize our results. In the revised manuscript, we have maintained the same droplet composition for comparison: a 0.4 μL droplet, consisting of 3.0 wt % dextran, 1.0 wt % PEO with a volume ratio of 9:1, and 1.0 wt % Pluronic F-68. The updated results are shown in the figure below (revised Figure 3). A circle of satellite microdroplets formed during evaporation for the organics-wetted case, but not for the unwetted case. The results and

conclusion remain the same as in the previous version. Follow the reviewer's suggestion, we have included the updated results in the revised manuscript.

(3) In response to the role of Pluronic F-68 in case (A), we have provided detailed explanation in the response to Comment 2.

Revised Figure 3. Morphology of an evaporating ATPS droplet (without proteins) resulting from segregative LLPS between PEO and dextran. Two cases are investigated: a sessile droplet evaporates on (A) an organics-wetted surface or (B) a clean surface. Schematic and corresponding confocal images of the resulting morphology after evaporation are shown. Dextran and PEO-rich phases are labeled by fluorescein isothiocyanate-dextran (FITC-dextran, green) and rhodamine B-polyethylene glycol (Rhodamine B-PEG, red), respectively. The sample was prepared by mixing 3.0 wt % dextran and 1.0 wt % PEO with a volume ratio of 9:1. At other volume ratios, a circle of satellite microdroplets were also visible (Figure S6). 0.4 μL of the sample was pipetted onto a petri dish where 300 μL silicone oil was used to wet the substrate *a priori* for the organics-wetted case. At all samples, 1.0 wt % Pluronic F-68 was contained. All scale bars are 200 μm .

Comment 5: Please clearly indicate which experiment used RSN-0749 or Pluronic F-68 and explain the reasons for using these additives.

Response: We thank the reviewer for the helpful suggestion. The reasons for using RSN-0749 or Pluronic F-68 were given in the response to Comment 2. Following the reviewer's advice, we have provided clear specification on the compositions of chemicals in each experiment in the revised manuscript. These specifications have been added to the captions of each figure, specifically,

(1) In the caption of Figure 4, on Page 12, Line 302-308, the following description is added: In case (A) and (B), a sessile droplet with a volume of 0.4 μL was placed on a 300 μL oil phase in the petri dish. This droplet contained 1.0 wt % Pluronic F-68 and 10.0 wt % glucose. We used 0.1 wt % purple-fluorescent PS beads as tracing particles, with diameters of 1 μm and 5 μm in case (A) and (B), respectively. In case (B), the oil was labelled by blue-fluorescent perylene. In case (C), the composition of the sessile droplet was the same as in Figure 2A.

(2) In the caption of Figure 5, on Page 14, Line 343-347, the following description

is added: In experiments, the sample was prepared by mixing 3.0 wt % dextran and 1.0 wt % PEO with a volume ratio of 1:1, and was added with 0.5 wt % lactoferrin, 0.5 wt % ovalbumin, and 0.2 mM/L RNA. Then, a 0.4 μ L sample droplet was placed on a 300 μ L oil phase (containing 8.0 wt % RSN-0749) in the petri dish. The RNA was labeled by Cy5-Poly U15 (red fluorescence).

Comment 6: It is also noted that the composition of the droplets (the concentration and the volume ratio of dextran to PEO) were different in each Figure. Please explain why.

Response: We thank the reviewer for pointing out this important issue. In the revised manuscript, we followed the reviewer's advice and maintained the concentrations of dextran and PEO at 3.0 wt % and 1.0 wt %, respectively, throughout all experiments, while varying the volume ratio.

The ATPS droplets used in experiments fall into two categories: those containing proteins (as in Figures 2 and 5), and those without proteins (as in Figures 3A and 4C). In the revised manuscript, we maintained the same volume ratio of dextran and PEO for experiments in the same category to make a fair comparison. For instance, in the cases with proteins, the volume ratio of dextran to PEO was 1:1. In the cases without proteins, this ratio was adjusted to 9:1 in the revised manuscript. The reason for not maintaining a 1:1 volume ratio in the latter case is that the droplet morphology was most distinct at a 9:1 ratio. We further explored the influence of various dextran and PEO compositions on the droplet morphology during evaporation. Results are included in Figure S6 of the revised SI, where a circle of satellite microdroplets can form in the organics-wetted case regardless of the droplet composition.

To address the reviewer's comments, we have made necessary changes and included the updated results in the revised manuscript on Page 9, Line 223-227.

Figure S6. Morphology of ATPS droplets evaporating on organics-wetted and clean substrates at different volume ratios from 1:1 to 14:1. For organics-wetted case, 300 μ L silicone oil was used to wet the substrate *a priori*. At all samples, 1.0 wt % Pluronic F-68 was contained. All scale bars are 200 μ m.

Response to Reviewer #2:

Comment: The authors have addressed my principal concern with the first submission. I believe that this version is suitable for publication in Nature Communications.

Answer: We thank the reviewer for the positive feedback. We are glad that our revisions have addressed the reviewer's concerns, and we are grateful for his/her endorsement to publish in Nature Communications.

Response to Reviewer #3:

Comment 1: The manuscript titled “Multicompartmental coacervate-based protocell by spontaneous droplet evaporation” authored by Cheng Qi et. al. describes the formation of core-shell coacervate based protocells driven by physical flows from an evaporating droplet. The authors have undertaken additional experiments to address the questions that I had about their previous manuscript. The additional experiments have provided more evidence to support their arguments.

Response: We appreciate the reviewer’s acknowledgement of our additional experiments and efforts. We are grateful for their recognition of the enhanced evidence supporting our arguments.

Comment 2: With the additional control experiments they show that a major factor for preventing droplet coalescence is due to the oil. This suggests that the balance of flows driven by evaporation of the droplet is not the dominating factor for the stabilizing the core shell coacervate structures. That the stabilization is a thermodynamic rather than a kinetic effect. If this is the case, the manuscript should be revised accordingly to:

- More clearly describe the “organics-wetted surface” in the abstract and introduction (for example lines 64-68)
- Clearly describe the factors which are important for the droplet stabilization
- Describe the context of the maragoni and capillary effects in light of the new experiments.

Response: We thank the reviewer for his/her constructive comments. The reviewer is right that the oil film coverage plays a crucial role in stabilizing the droplets. Specifically, a partial oil film coverage on the water droplet create a circle of stagnation points near the water-oil-air contact line. Microdroplets trapped at these stagnation points are prevented from coalescence. Following the reviewer’s suggestion, we have revised the manuscript to more clearly describe the “organics-wetted surface” in the abstract and introduction on Page 1, Line 24-30, and Page 2, Line 60-61. Additionally, we have clearly discussed the factors critical for droplet stabilization, at Page 3, Line 75-79. Moreover, we have described the Marangoni and capillary effects in accordance with the new experimental data on Page 8, Line 213-214 in the revised manuscript.

Comment 3: The authors say that the added value of their work is that they show the formation of hierarchical structures in “natural conditions”. It can be argued that a simulated surface with oil on it is not natural and is a laboratory mimic for a potential real-life scenario. As such, it does not set this work above other studies which show hierarchical structuration in coacervates in laboratory scenarios. The text should be modified to focus on the phenomena that drives and stabilizes the droplets rather than the “natural conditions”.

Response: We appreciate the reviewer's insightful comment, which has helped us better emphasize the significance of our work. Our experiments indeed mimic potential real-life scenario. In response to reviewer’s suggestion, we have revised our manuscript to focus more on the mechanisms that create and stabilize the droplets, rather than on the

concept of "natural conditions". You can find these significant alterations on Page 1, Line 28-30; Page 3, 78-79; Page 3, Line 83-86; Page 14, Line 354; Page 15, Line 381-382 in the updated manuscript.

Comment 4: Line 96. They say that this work shows the “potential origins of complex architectural structures under natural conditions”. This sentence should be justified in light of a number of papers which show the formation of multistructures under different conditions.

Response: We thank the reviewer for the helpful suggestion. We revised such sentences on Page 3, Line 83-86; Page 14, Line 354; and Page 15, Line 381-382 in the revised manuscript.

Comment 5: Language can be improved in various places.

Response: We appreciate the reviewer's feedback. In response to their comment, we have thoroughly reviewed the manuscript and made the necessary language improvements, which are highlighted in red.

REVIEWERS' COMMENTS

Reviewer #1 (Remarks to the Author):

The authors have addressed my concerns seriously and revised the manuscript accordingly. I think this version of manuscript could be published in Nature Communications.

Reviewer #3 (Remarks to the Author):

I appreciate the efforts in addressing the previous comments regarding the manuscript, however, there are still instances where the comments have not been fully addressed. Please see below:

Comment 2: With the additional control experiments they show that a major factor for preventing droplet coalescence is due to the oil. This suggests that the balance of flows driven by evaporation of the droplet is not the dominating factor for the stabilizing the core shell coacervate structures. That the stabilization is a thermodynamic rather than a kinetic effect. If this is the case, the manuscript should be revised accordingly to:

- More clearly describe the “organics-wetted surface” in the abstract and introduction (for example lines 64-68)
- Clearly describe the factors which are important for the droplet stabilization
- Describe the context of the maragoni and capillary effects in light of the new experiments.

Response: We thank the reviewer for his/her constructive comments. The reviewer is right that the oil film coverage plays a crucial role in stabilizing the droplets. Specifically, a partial oil film coverage on the water droplet create a circle of stagnation points near the water-oil-air contact line. Microdroplets trapped at these stagnation points are prevented from coalescence. Following the reviewer’s suggestion, we have revised the manuscript to more clearly describe the “organics-wetted surface” in the abstract and introduction on Page 1, Line 24-30, and Page 2, Line 60-61. Additionally, we have clearly discussed the factors critical for droplet stabilization, at Page 3, Line 75-79. Moreover, we have described the Marangoni and capillary effects in accordance with the new experimental data on Page 8, Line 213-214 in the revised manuscript.

Reviewer: The authors have focused here on the change of Marangoni and capillary effects as well as the evaporation effects as a consequence of the oil film upon the surface. How about droplet stabilisation from the oil? It seems that this could play a role in preventing droplet coalescence. It would be important to add a comment about this.

Comment 4: Line 96. They say that this work shows the “potential origins of complex architectural structures under natural conditions”. This sentence should be justified in light of a number of papers which show the formation of multistructures under different conditions.

Response: We thank the reviewer for the helpful suggestion. We revised such sentences on Page 3, Line 83-86; Page 14, Line 354; and Page 15, Line 381-382 in the revised manuscript.

Reviewer: The authors may have misunderstood what was asked for here. Please add additional references of work, where there are multiple which show the formation of multistuctures under different conditions.

Comment 5: Language can be improved in various places.

Response: We appreciate the reviewer's feedback. In response to their comment, we have thoroughly reviewed the manuscript and made the necessary language improvements, which are highlighted in red.

Reviewer: There are still mistakes in the language. Perhaps the editor can also check through the language elements?

Response to Reviewer #1:

Comment: The authors have addressed my concerns seriously and revised the manuscript accordingly. I think this version of manuscript could be published in Nature Communications.

Answer: We thank the reviewer for the positive feedback. We are glad that our revisions have addressed the reviewer's concerns, and we are grateful for his/her endorsement to publish in Nature Communications.

Response to Reviewer #3:

Comment 2: With the additional control experiments they show that a major factor for preventing droplet coalescence is due to the oil. This suggests that the balance of flows driven by evaporation of the droplet is not the dominating factor for the stabilizing the core shell coacervate structures. That the stabilization is a thermodynamic rather than a kinetic effect. If this is the case, the manuscript should be revised accordingly to:

- More clearly describe the “organics-wetted surface” in the abstract and introduction (for example lines 64-68)
- Clearly describe the factors which are important for the droplet stabilization
- Describe the context of the marangoni and capillary effects in light of the new experiments.

Response: We thank the reviewer for his/her constructive comments. The reviewer is right that the oil film coverage plays a crucial role in stabilizing the droplets. Specifically, a partial oil film coverage on the water droplet create a circle of stagnation points near the water-oil-air contact line. Microdroplets trapped at these stagnation points are prevented from coalescence. Following the reviewer’s suggestion, we have revised the manuscript to more clearly describe the “organics-wetted surface” in the abstract and introduction on Page 1, Line 24-30, and Page 2, Line 60-61. Additionally, we have clearly discussed the factors critical for droplet stabilization, at Page 3, Line 75-79. Moreover, we have described the Marangoni and capillary effects in accordance with the new experimental data on Page 8, Line 213-214 in the revised manuscript.

Reviewer: The authors have focused here on the change of Marangoni and capillary effects as well as the evaporation effects as a consequence of the oil film upon the surface. How about droplet stabilisation from the oil? It seems that this could play a role in preventing droplet coalescence. It would be important to add a comment about this.

Response: We thank the reviewer for the comment. In the revised manuscript, to emphasize the importance of the oil, we add a comment on how droplets are stabilized from the oil in the Abstract, on Page 1, Line 26-28.

Comment 4: Line 96. They say that this work shows the “potential origins of complex architectural structures under natural conditions”. This sentence should be justified in light of a number of papers which show the formation of multistructures under different conditions.

Response: We thank the reviewer for the helpful suggestion. We revised such sentences on Page 3, Line 83-86; Page 14, Line 354; and Page 15, Line 381-382 in the revised manuscript.

Reviewer: The authors may have misunderstood what was asked for here. Please add additional references of work, where there are multiple which show the formation of multistructures under different conditions.

Response: We thank the reviewer for the suggestion. We additionally cited the following studies for the sentence “*Maintaining the hierarchical structure of coacervate-based protocells is essential for their function as organization hubs*” on Page 2, Line 44-45:

- Rai, S. K. et al. Heterotypic electrostatic interactions control complex phase separation of tau and prion into multiphasic condensates and co-aggregates. *Proceedings of the National Academy of Sciences* 120, e2216338120 (2023).
- Liu, X. et al. Multiphasic Coacervates Assembled by Hydrogen Bonding and Hydrophobic Interactions. *J Am Chem Soc* 145, 23109–23120 (2023).
- Donau, C. et al. Phase Transitions in Chemically Fueled, Multiphase Complex Coacervate Droplets. *Angew Chem Int Edit* 61, e202211905 (2022).
- Ye, S. et al. Micropolarity governs the structural organization of biomolecular condensates. *Nat Chem Biol* (2023) doi:10.1038/s41589-023-01477-1.
- Mu, W. et al. Superstructural ordering in self-sorting coacervate-based protocell networks. *Nat Chem* (2023) doi:10.1038/s41557-023-01356-1.

Comment 5: Language can be improved in various places.

Response: We appreciate the reviewer's feedback. In response to their comment, we have thoroughly reviewed the manuscript and made the necessary language improvements, which are highlighted in red.

Reviewer: There are still mistakes in the language. Perhaps the editor can also check through the language elements?

Response: We thank the reviewer for his/her careful review. We scrutinize the manuscript to ensure that there are no spelling and grammatical mistakes.